# Regional New Particle Formation as Modulators of Cloud Condensation Nuclei and Cloud Droplet Number in the Eastern Mediterranean

**Panayiotis Kalkavouras[1,2], Aikaterini Bougiatioti[1,2], Nikos Kalivitis[1], Iasonas Stavroulas[1,2,3], Maria Tombrou[4], Athanasios Nenes[5,2,6], and Nikolaos Mihalopoulos[1,2]**

[1]Enviromental Chemical Processes Laboratory, Department of Chemistry, University of Crete, Heraklion, 71003, Greece
[2]Institute of Environmental Research & Sustainable Development, National Observatory of Athens, Palea Penteli, 15236, Greece
[3]Energy Environment and Water Research Center, The Cyprus Institute, Nicosia 2121, Cyprus
[4]Department of Physics, University of Athens, Athens, 15784, Greece
[5]Laboratory of Atmospheric Processes and their Impacts, School of Architecture, Civil & Environmental Engineering, École Polytechnique Fédérale de Lausanne, 1015, Lausanne, Switzerland
[6]Institute for Chemical Engineering Science, Foundation for Research and Technology Hellas, Patras, 26504, Greece

*Correspondence to*: athanasios.nenes@epfl.ch; abougiat@noa.gr

## Abstract

A significant fraction of atmospheric particles that serve as cloud condensation nuclei (CCN) are thought to originate from the condensational growth of new particles formed (NPF) from the gas phase. Here, 7 years of continuous aerosol and meteorological measurements (June 2008 to May 2015) at a remote background site of the eastern Mediterranean were recorded and analyzed to assess the impact of NPF (of 162 episodes identified) on CCN and cloud droplet number concentration (CDNC) formation in the region. A new metric is introduced to quantitatively determine the initiation and duration of the influence of NPF on the CCN spectrum. NPF days were found to increase CCN concentrations (between 0.10 and 1.00% supersaturation) between 29 and 77%. Enhanced CCN concentrations from NPF are mostly observed, as expected, under low pre-existing particle concentrations, and occur in the afternoon, relatively later in the winter and autumn than in the summer. Potential impacts of NPF on cloud formation was quantified by introducing the observed aerosol size distributions and chemical composition into an established cloud droplet parameterization. We find that the supersaturations that develop are very low (ranging between 0.03 and 0.271%) for typical boundary layer dynamics ($\sigma_w \sim 0.3$ m s$^{-1}$) and NPF is found to enhance CDNC by a modest 13%. This considerable contrast between CCN and CDNC response is in part from the different supersaturation levels considered, but also because supersaturation drops from

increasing CCN because of water vapor competition effects during the process of droplet formation. The low cloud supersaturation further delays the appearance of NPF impacts on CDNC to clouds formed in the late evening and nighttime – which carries important implications for the extent and types of indirect effects induced by NPF events. An analysis based on CCN concentrations using prescribed supersaturation can provide much different,

and even misleading, conclusions and should therefore be avoided. The proposed approach here offers a simple, yet highly effective way for a more realistic impact assessment of NPF events on cloud formation.

## 1. Introduction

Cloud condensation nuclei (CCN) and cloud droplet formation constitutes the direct microphysical link between aerosols and clouds. Quantifying how changes in aerosols affect global clouds, precipitation and climate is limited by the large number of processes and scales that need to be captured in models (Stevens and Feingold, 2009; Pöschl et al., 2010; Seinfeld et al., 2016; Cecchini et al., 2017). New particle formation (NPF), the process during which

new particles are formed directly from the gas-phase, is thought to significantly shape the distribution of CCN throughout the atmosphere (Pierce and Adams 2007; Westervelt et al., 2013; Gordon et al., 2017). Although initially too small (1–2 nm; Kerminen et al., 2012) to act as CCN, particles from NPF can grow to sufficient size and hygroscopicity over a period of few hours to days and eventually act as efficient CCN.

Field studies have demonstrated substantial local enhancement in CCN number from NPF. For example, Wiedensohler et al. (2009) observed that the CCN size distribution was dominated by the growing nucleation-mode (above 80%) in a highly polluted region around Beijing, while Dameto de España et al. (2017), found that NPF in Vienna, Austria increases the CCN number concentration by up to 143% at 0.50% supersaturation. Sihto et al. (2011)

found at the Hyytiälä Forestry Field Station of the University of Helsinki, Finland that NPF increases the CCN concentrations in the evening of a NPF day by 70-110% depending on the supersaturation level, while Rose et al. (2017) observed that CCN concentrations were increased by 168 to 996% at Chacaltaya during NPF events at the Chacaltaya station, Bolivia (5,240 m a.s.l.). Additionally, model investigations suggest atmospheric NPF to be an

important contributor to CCN, and thereby to aerosol-cloud-climate interactions. Spracklen et al. (2008) have shown that boundary layer (BL) particle formation can cause an increase in global BL CCN concentrations at 0.20% supersaturation by 3-20%, and by 5-50% at 1.00% supersaturation, respectively. Merikanto et al. (2010) found that 45% of global low-level cloud CCN at 0.20% supersaturation originates from NPF. Moreover, Westervelt et al. (2014)

estimated an average 49 - 78% increase in global boundary-layer CCN number concentration (at 0.20% supersaturation) from NPF.

NPF events followed by growth to CCN-sized particles are observed to take place frequently and over relatively large spatial scales in continental boundary layers, including forested areas at mid and high latitudes, other remote continental regions, urban areas and even highly-
polluted environments (e.g. Kerminen et al., 2018). NPF events are long known to occur in marine environments, highlighting the role of iodine species as precursors for new particle cluster formation (Sellegri et al., 2016), from oxidation of biogenic alkyl-halides in near-coastal areas (e.g. O'Dowd et al., 2002; Vaattovaara et al., 2006) and providing the most comprehensive mechanistic description of coastal NPF presented to date (Sipilä et al., 2016).
Furthermore, NPF can be triggered by the rapid dimethylsulphide (DMS) oxidation above clouds (Bates et al., 1987; Kreidenweis et al., 1991; Katoshevski et al., 1999) and cloud outflow regions associated with convection (e.g. Hermann et al., 2003). NPF within marine boundary layers can strongly affect CCN number concentrations at all cloud-relevant supersaturations (e.g. Kalivitis et al., 2015; Kalkavouras et al., 2017; Debevec et al., 2018).
When these small particles however are mixed within the boundary layer, they may subsequently grow to CCN-relevant sizes, or even act as CCN in strongly convective clouds (Fan et al., 2013; Wang et al., 2016).

A thorough assessment of NPF impacts on CCN levels requires knowledge of all events and subsequent microphysical processing that occurred throughout the path of an air-mass.
Observationally, this is almost impossible to carry out; one can therefore only quantify the CCN concentration perturbation, or enhancement, above "background" levels that existed prior to an NPF event (Peng et al., 2014; Wu et al., 2015; Ma et al., 2016). Although conceptually straightforward, studies differ in the approach used to define the initiation of a NPF event (e.g. a strong enhancement in total particle number, the shape of the size
distribution), the pre-event CCN concentration (e.g. a 30-minute or 1 hour-average CCN concentration before the initiation time), and also the metric used to quantify the CCN enhancement from a NPF event (e.g. peak enhancement, a time-averaged enhancement, and the size defining the lower limit of CCN activation). Furthermore, observational studies quantify CCN enhancements from measurements of aerosol number size distribution; the link
to CCN concentrations is done by using a prescribed (or calculated) "critical diameter" ($d_c$), above which all particles act as CCN in clouds. Studies widely vary in the approach used to determine this critical diameter, $d_c$, so additional considerations are required between assessments. Theoretically, $d_c$ depends on the level of supersaturation that develops in clouds and the chemical composition of the particles (Seinfeld and Pandis, 2006). Often, $d_c$ is
prescribed between 50 and 150 nm, corresponding roughly to clouds with maximum

saturation levels between 1.00%, and 0.10%, respectively (Kerminen et al. 2012). However, clouds are not characterized by a constant supersaturation, rather exhibit variable levels that instantaneously adjust to the intensity of cloud updrafts and the CCN spectra (e.g. Nenes and Seinfeld, 2003; Hudson et al., 2014). It is clear that all the above conventions need careful consideration, as they can affect the magnitude and duration of CCN enhancement for each event.

Asmi et al. (2011) at the Pallas GAW station in northern Finland estimated the contribution of NPF to CCN concentration. The method adopted was to subtract the concentration of particles larger than 80 nm diameter ($N_{80}$) at the end of the NPF, from the average $N_{80}$ before the NPF influence (defined from the time where the NPF started up to where the nucleation-mode particles reach 80 nm diameter). A similar approach was used to quantify the enhancement from NPF to particles larger than 50, and 100 nm ($N_{50}$, and $N_{100}$, respectively). The relative enhancement of $N_{50}$, $N_{80}$, and $N_{100}$ from NPF was $160\pm270\%$, $210\pm110\%$, and $50\pm130\%$, respectively. In the boreal forest station of Hyytiälä, Kerminen et al. (2012) calculated the CCN number concentrations using the particle number size distributions, for diameters above 50, and 100 nm. The contribution of any NPF event was determined from the comparison of the maximum particle number concentration ($N_{max}$) that develops during an event (1-h average) over the particle number concentration ($N_{prior}$) prior to the event (1-h average). $N_{50(max)}$ / $N_{50(prior)}$ and $N_{100(max)}$ / $N_{100(prior)}$ presented an increase of 317% and 202%, respectively, in CCN concentration. The approach of Kerminen et al. (2012) has been used in China (Peng et al., 2014), where the contribution of NPF events to CCN at 0.20% supersaturation was 6% on regional sites, while Wu et al. (2015) using 2-h averaging in Melpitz, Germany found that NPF enhance CCN number concentration 63, 66, and 69% for 0.10, 0.40, and 0.60% supersaturation, respectively.

Apart from impacting CCN number concentrations, NPF events can also influence clouds and climate by promoting cloud dimming, thus regional warming, during periods with high NPF frequency over the comparatively polluted area of Midwestern USA (Sullivan et al., 2018). Furthermore, it is clear that the timing of the initiation of the NPF event and the subsequent growth of particles to CCN and eventually droplets is of utmost importance, as the time delay between the different processes actually limits the time during which the albedo of clouds is affected by NPF. In reality, the total contribution of atmospheric nucleation (including indirect effects) to a net short-wave radiation balance in the atmosphere, depends on the rate in which the emissions of gas-phase compounds are responsible for nucleation (and subsequent growth), as well as of primary particles, act as a sink for the freshly formed particles, during a NPF day.

Although most prior observation studies linked NPF to CCN number enhancement, very few actually link NPF to the process of cloud droplet formation and cloud droplet number concentration (CDNC). The latter distinction is important, given that droplet number in clouds exhibit a sub-linear response to aerosol increases (Twomey et al., 1977; Leaitch et al., 1986; Ghan et al., 1993; Boucher and Lohmann, 1995; Gultepe and Isaac, 1996; Nenes et al., 2001; Ramanathan et al., 2001; Ghan et al., 2011; Sullivan et al., 2016), owing to the elevated competition for water vapor and reduction in cloud supersaturation. The understanding of NPF impacts on CCN levels may therefore provide a biased view on its potential impact on droplet number ($N_d$) and the aerosol indirect effect. Using cloud droplet parameterizations to interpret observed aerosol size distribution data, however, may allow one to address this issue in a simple but effective way. Kalkavouras et al. (2017) illustrated this issue by using a "conventional" approach to quantify CCN enhancement, using the critical diameter ($d_c$) at which all particles act as CCN depending on observed composition and a prescribed supersaturation. They reported much higher CCN number enhancement (~87%) for two sites in the eastern Mediterranean (Santorini and Finokalia) than in cloud droplet number concentration, $N_d$, (~12%) during two consecutive NPF episodes in summer. The reason for this 8-fold discrepancy lies in the drastically different supersaturation used to quantify CCN enhancement (0.20, 0.40, 0.60, and 0.80%) than what was computed for cloud droplet number concentrations (0.10 and 0.13% for updraft velocities of 0.3 m s$^{-1}$ and of 0.6 m s$^{-1}$, respectively).

The current study follows up on the initial work of Kalkavouras et al. (2017) and quantifies the impact of NPF on CCN levels and cloud droplet number concentrations in the eastern Mediterranean atmosphere over 7 years of field measurements (June 2008 to May 2015) of aerosol number size distributions and chemical composition. From this data, we aim to (*i*) quantify the seasonality and contribution of atmospheric NPF to the production of newly CCN in the eastern Mediterranean marine atmosphere, (*ii*) determine the timing properties of newly-formed particles from the beginning of NPF events (i.e. starting time ($t_{start}$) and duration) throughout their activation into cloud droplets, and their relative contribution to the CCN budget, and, (*iii*) investigate the NPF impacts on CDNC ($N_d$) and on maximum supersaturation ($s_{max}$) formed in clouds in the vicinity of Finokalia. In the process of addressing these goals, we consider major issues regarding the calculation of cloud supersaturation and event characteristics that affect the NPF impact calculations.

## 2. Methodology

## 2.1 Experimental site

From June 2008 to May 2015, measurements were performed at the atmospheric observation station of the University of Crete at Finokalia, Crete, Greece (35º 20′ N, 25º 40′ E; 50 m from

the shore and 250 m a.s.l.). The monitoring station of Finokalia (http://finokalia.chemistry.uoc.gr/) is located at the top of a hill over the coastline, in the northeast part of the island of Crete, facing the Aegean Sea in the wide north sector. Since the site was established in 1993, Finokalia experiences two characteristic periods during the year; the dry period from April to September, and the wet one from October to March. The dry period is dominated by strong winds of N/NW direction (up to 90%, originating from central and eastern Europe and Balkans) of speed exceeding 10 m s$^{-1}$. The wet period is characterized by limited prevalence of the N/NW sector, and significant transport from Sahara (S/SW winds; occurrence up to 20%). An extensive description of the site and prevailing meteorology can be found in Mihalopoulos et al. (1997).

## 2.2 Aerosol composition and size distribution

Number size distribution of particles having mobility diameters from 9 to 848 nm (scanned range) were measured with a 5 min time resolution, using a custom-built scanning mobility particle sizer (SMPS; TROPOS-Type, Wiedensohler et al., 2012). The system is a closed-loop, with a 5:1 ratio between the aerosol and sheath flow, and it comprises a Kr-85 aerosol neutralizer (TSI 3077), a Hauke medium differential mobility analyzer (DMA), and a TSI-3772 condensation particle counter (CPC). The sampling was made through a PM$_{10}$ sampling head and the sample humidity was regulated to a relative humidity below 40% using Nafion® dryers in both the aerosol and sheath flow. Particles were charged via a Kr-85 neutralizer, and thereafter introduced into the DMA. By setting different voltages in the DMA, particles of different electrical mobility are selected and their particle number concentration can be measured. The fluctuation of voltage yields an electrical particle mobility distribution, which can be inverted into a particle number size distribution. The recorded number size distributions were corrected for particle losses by diffusion on the various parts of the SMPS following the recommendations by Wiedensohler et al. (2012). Three different types of calibration were performed for the SMPS, namely DMA voltage supply calibration, aerosol and sheath flows calibrations, and size calibrations.

The complete dataset of particle size distributions was checked for the presence of NPF events, identified by a sudden increase of the nucleation-mode particles concentration (i.e. those with diameters below 25 nm), and further growth of these freshly-formed particles that lead to a continuous increase in larger particle concentrations over a short period of time (usually less than 4h) (Kulmala et al., 2004). The NPF event progression is characterized by the relative changes of the three particle-modes, "nucleation" (diameter less than 25 nm), "Aitken" (diameter between 25 and 100 nm), and "accumulation" (diameter larger than 100 nm). The modal concentration of particles is obtained from the respective SMPS size bins, as follows:

$$N_{\text{nucleation}} = \int_0^{25} n(d_p) \, \mathrm{d}d_p \approx \sum\nolimits_9^{25} \Delta N_i \tag{1}$$

$$N_{\text{Aitken}} = \int_{25}^{100} n(d_p) \, \mathrm{d}d_p \approx \sum\nolimits_{25}^{100} \Delta N_i \tag{2}$$

$$N_{\text{Accumulation}} = \int_{100}^{\infty} n(d_p) \, \mathrm{d}d_p \approx \sum\nolimits_{100}^{848} \Delta N_i \tag{3}$$

where $n(d_p)$ is the aerosol number size distribution, $\Delta N_i$ is its binned approximation from the SMPS data for particles in each mode (9-25 nm for nucleation, 25-100 nm for Aitken, and 100-848 nm for accumulation) and particle concentration of each mode being the sum of particle concentration in all size bins of the corresponding diameter range. The upper and

lower sizes are limits of size detection for the particular SMPS.

From the period between June 2008 and December 2011, the bulk aerosol chemical composition of $PM_{10}$ was measured in parallel with the size distributions using daily 24-h quartz fiber filters (PALL Tissuquartz, 2500 QAT 47 mm). Samples were analyzed for water-soluble ions after extraction with nanopure water. The solutions acquired were first filtered

using syringe filters (PALL IC Acrodisc® (PES), 0.45 μm, 13 mm) to remove any non-soluble species and subsequently analyzed using ion chromatography (IC) for anions ($Cl^-$, $Br^-$, $NO_3^-$, $SO_4^{2-}$) and cations ($K^+$, $Na^-$, $NH_4^+$, $Mg^{2+}$, $Ca^{2+}$), using the procedure of Bardouki et al. (2003). Furthermore, the $PM_{10}$ quartz filters were analyzed for organic and elemental carbon (Carbon Aerosol Analysis Lab Instrument, SUNSET Laboratory Inc.) using the

EUSAAR 2 protocol of analysis (Cavalli et al., 2010). For the estimation of the fine particulate matter fraction ($PM_1$) chemical composition, the respective concentrations of sulfates, organics, and ammonium from the bulk $PM_{10}$ are considered using the approach presented in Bougiatioti et al. (2009). According to this study, bulk chemical composition from daily filter analysis was used to calculate the volume fraction of organics and

ammonium sulfate. With the subsequent application of Köhler theory, CCN number concentrations were calculated for closure purposes considering two different scenarios for the solubility of organics. As far as CCN concentrations are concerned, results showed that limitations of using bulk, instead of size-resolved and daily chemical composition are minimal, as CCN closure was achieved with an error of $0.6\pm6\%$. For the conversion of

organic carbon to matter needed for the application of Köhler theory and the calculation of the organics volume fraction, a ratio of OM/OC of 2.1 was used, based on other studies from this site (Sciare et al., 2005; Hildebrandt et al., 2010). Any CCN prediction uncertainty from using bulk, daily chemical composition is further reduced when used to compute droplet number (e.g., Sotiropoulou et al., 2007; Kalkavouras et al., 2017).

From May 2012 to May 2015, the mass and chemical composition of non-refractory submicron aerosol particles ( $SO_4^{2-}$, $NO_3^-$, $NH_4^+$, $Cl^-$, and organic matter) was provided with a 30 min time resolution, by an Quadrupole Aerosol Chemical Speciation Monitor (ACSM), equipped with a standard vaporizer (Ng et al., 2011). The instrument sampled through a BGI Inc. SCC 1.197 sharp cut cyclone operated at 3 L min[-1], yielding a cut-off diameter of almost 2 μm. The response factor (RF) for nitrate along with the relative ionization efficiencies (RIEs) for ammonium and sulfate were determined by ammonium nitrate and ammonium sulfate calibrations, and the RIE for sulfate was determined according to the fitting approach proposed by Budisulistiorini et al. (2014). Mass concentrations were corrected using a chemical composition dependent collection efficiency (Middlebrook et al., 2012).

## 2.3 Cloud Condensation Nuclei (CCN)

Measurements of cloud condensation nuclei (CCN) concentration (cm[-3]) between 0.38 and 0.73% supersaturation were conducted using a Droplet Measurement Technologies (DMT) Continuous Flow Streamwise Thermal Gradient CCN counter (CFSTGC; Roberts and Nenes, 2005), from November 2014 to May 2015. The CFSTGC is composed of a cylindrical diffusion chamber in which supersaturation is generated and controlled by the air flow rate, pressure, and a streamwise temperature gradient maintained by a heater and a set of thermoelectric coolers (Roberts and Nenes, 2005; Lance et al., 2006). The airflow rate used was 0.5 L min[-1] with a sheath-to-aerosol flow ratio of 10:1, and a top-bottom column temperature difference, ΔT, between 4 and 15 K. Concentrations were measured at each supersaturation (0.38, 0.52, 0.66, and 0.73%) for 15 min, yielding a CCN spectrum consisting of 4 different supersaturations approximately every hour. Calibration of the instrument supersaturation was performed by determining the minimum diameter of monodisperse ammonium sulfate aerosol generated from a differential mobility analyzer (DMA), which activates at given chamber flow rate, ΔT, and chamber pressure following the procedure of Bougiatioti et al. (2009). The CCN instrument was calibrated numerous times throughout the campaign. For the lower supersaturation, the relative variability between calibrations did not exceed 1%, whereas for the highest supersaturation the variability was under 4%. As CCN concentrations during the measurement period rarely exceeded 5,000 cm[-3], no correction for water vapor depletion inside the CFTGC chamber was deemed necessary (Lathem and Nenes, 2011).

## 2.4 Calculation of CCN concentrations from size distribution data

As in numerous prior studies, CCN number concentrations can be calculated from the observed number size distributions by integrating the SMPS data from a characteristic diameter $d_c$ to the largest size particles measured:

$$\mathrm{CCN}(d_\mathrm{c}) = \int_{d_c}^{\infty} n(d_p)\mathrm{d}d_p \approx \sum \ _{dc}^{848} \Delta N_i \tag{4}$$

where $d_c$ is the SMPS size bin that contains the critical diameter and 848is the bin with the largest particles measured by the SMPS. Instead of prescribing $d_c$ (as done in other studies), we link it to a desired supersaturation level, $s_\mathrm{c}$, using $\kappa$-Köhler theory:

$$d_c = \left(\frac{4A^3}{27\kappa s_c^2}\right)^{1/3} , \mathrm{A} = \frac{4M_\mathrm{w}\sigma_\mathrm{w}}{\mathrm{R}T\rho_\mathrm{w}} , \tag{5}$$

where $\mathrm{M_w}$ is the molar mass of water, $\sigma_\mathrm{w}$ is the surface tension of water, R is the universal gas constant, T is the temperature, and $\rho_\mathrm{w}$ is the density of water. Even though when using bulk, daily chemical composition, one kappa value is used per day, $d_c$ changes also depend on temperature and critical supersaturation. In our case, where past experience has shown that the composition displays remarkably consistent behavior (Bougiatioti et al., 2009; 2011) the successful CCN closure shows that indeed the used approach is sufficient in calculating effectively the $d_c$ and not using a prescribed value. CCN number concentrations are then taken as being equal to the concentration of particles with diameter above $d_c$ (Kalkavouras et al., 2017). The aerosol hygroscopicity parameter, $\kappa$, is calculated assuming that it is a mixture of an organic and inorganic component with volume fraction $\varepsilon_{org}$, $\varepsilon_{inorg}$ and characteristic hygroscopicity $\kappa_{org}$, $\kappa_{inorg}$ respectively ($\kappa = \varepsilon_{inorg}\kappa_{inorg} + \varepsilon_{org}\kappa_{org}$). Past studies at Finokalia have suggested that prescribing $\kappa_{org} = 0.16$ and $\kappa_{inorg} = 0.6$ reproduces CCN to within 2% on average, but exhibit some size dependence (Bougiatioti et al., 2009; 2011). Furthermore, Koulouri et al. (2008), and Bougiatioti et al. (2013) have established that sulfate is by majority found in the fine fraction (82.7±12.7% of $PM_{10}$ sulfate found in $PM_1$) and the same applies also for ammonium (88±13.3% of $PM_{10}$ ammonium found in $PM_1$). Therefore the uncertainty on the $\kappa$ calculation, as far as sulfate and ammonium is concerned arising from the use of $PM_{10}$ chemical composition to derive the respective $PM_1$ information is minimal. This is not the case for the organic matter, as it appears that 75±11% of $PM_{10}$ organic matter is found in $PM_1$. This is translated in a difference in the calculation of $\kappa$ in the order of 2.5±0.2%, with the recalculated $\kappa$ values being higher, as organics contribution decreases. Nevertheless, this 2.5% difference in kappa has an almost insignificant impact on CDNC and CCN, as changes of kappa by more than a factor of 2 are expected to begin impacting on CDNCs.

Indicatively, for 4 NPF days during August and September 2012, the combined processing of the concurrent CCN and ACSM data during NPF events provides the size-resolved $\kappa$ (Fig. S1), which can be used to assess the validity of using a common $\kappa$ for all sizes (supersaturations). For supersaturations below 0.20%, the size-resolved $\kappa$ from the CCN data is higher by 23% compared to the bulk $\kappa$ from the ACSM data, while for supersaturations between 0.20 and 0.40%, the CCN-derived values agree quite well with bulk chemical

composition data (slope 0.94), but with considerable scatter. For supersaturations above
0.40%, $\kappa$ derived from the chemical composition data exhibits on average an overestimation
bias of 38.5%. Altogether, the $\kappa$ trends suggest that the composition of particles tends to
increasingly deviate (or vary) from the bulk as they get smaller (i.e. with higher
supersaturation) – indication of enrichment by organics, often observed for NPF-derived
particles (e.g. Cerully et al., 2011). The large scatter at around 0.40% supersaturation can be
attributed to chemical composition fluctuations, given that concentrations are affected by both
the fresh organic-rich and aged sulfate-rich modes, more at least than found in the higher or
lower supersaturation CCN. Overall however, this level of hygroscopicity error is not
expected to induce substantial errors in CCN concentration predictions, as demonstrated in
the closure study below; a size-dependent consideration of hygroscopicity is therefore deemed
unnecessary.

We subsequently test the aforementioned approach for calculating CCN from chemical
composition and size-distribution measurements (Eq. 4) against direct CCN measurements
(Section 2.3) collected from November 2014 to May 2015. The degree of "CCN closure" is
assessed with 5 minute-averaged data at 0.38, 0.52, 0.66, and 0.73% supersaturation (Fig. S2).
The measured values of CCN at each supersaturation correlate strongly with the predicted
values, when considering all the available data. With increasing supersaturation, $s$, the value
of the coefficient of determination ($R^2$) increased and the scattering of data decreased (Table
S1). For the lowest supersaturations (0.38 and 0.52%), there is an overestimation (22%) of
predicted CCN concentrations – consistent with the fact that using bulk $\kappa$, which is higher
than the "real" size-dependent $\kappa$, would lead to slight overestimations in CCN. Interestingly
enough, although these $\kappa$ biases increase with decreasing size, the overestimation and scatter
in CCN is decreased, for the higher supersaturations (0.66 and 0.73% - estimated and
measured values agree within 10%) because an increasingly larger fraction of the aerosol
activates, so the error in absolute CCN number is diminished. Regardless of supersaturation,
CCN prediction errors and scatter do not seem to exceed 40%; these are considered minor,
especially within the context of droplet number calculations – because the former exhibit a
strongly sub-linear response to CCN changes in the eastern Mediterranean (e.g. Bougiatioti et
al., 2016; Kalkavouras et al., 2017) which means that CCN errors translate to much smaller
errors in CDNC. Conversely, to contrast our method against using a prescribed $d_c$, from the
available CCN data we calculated a mean $d_c$ at each supersaturation level, and afterwards
estimated the CCN number concentrations for this respective "fixed" $d_c$. Using both the
calculated CCN from a "fixed" $d_c$ against the CCN concentrations from chemical composition
and size-distribution measurements, we evaluated the two different approaches at 0.20, 0.38,
0.52, 0.73 and 1.00% suprsaturation, respectively. The values of our initial approach with

estimated CCN concentrations from kappa and size-distribution measurements, are generally
higher. More specifically, when using a "fixed" $d_c$ estimated CCN concentrations are almost
30% lower compared to the respective ones when using kappa and size-distribution
measurements for all supersaturation levels above 0.38%, whereas for 0.20% supersaturation,
the estimated CCN concentrations are approximately 60% lower for the "fixed" $d_c$ approach,
and this would further translate in higher discrepancies in an attempted closure study.

**2.5 Cloud droplet formation calculations**

From knowledge of the aerosol hygroscopicity, size distribution and cloud updraft velocity,
we can determine the cloud droplet number concentrations ($N_d$) and maximum supersaturation
for clouds forming in the vicinity of Finokalia, during all NPF events. Such calculations are
useful to directly link aerosol with CDNC in NPF-influenced clouds, and determine the
"cloud-relevant" supersaturations for which CCN perturbation calculations are relevant. For
such calculations we use the droplet parameterization based on the "population splitting"
concept of Nenes and Seinfeld (2003), later improved by Fountoukis and Nenes (2005),
Barahona et al. (2010), and Morales and Nenes (2014). These formulations provide a rapid
and accurate calculation of CDNC that forms in cloud updrafts, and largely captures the
CDNCs that form in ambient clouds (e.g. Ghan et al., 2011; Morales et al., 2011). When
calculating $N_d$, the size distribution is described using a sectional representation (Nenes and
Seinfeld, 2003) derived directly from the SMPS distribution data, similar to what was done in
Kalkavouras et al. (2017). Observations of cloud updraft velocity are not available at
Finokalia for the time period examined, but published measurements and model simulations
suggest that the distribution of updraft velocities in cloudy boundary layers in the region of
Finokalia show a dispersion of $\sigma_w = 0.2\text{-}0.3$ m s$^{-1}$ during the period of northerly (Etesian)
winds (Tombrou et al., 2015; Dandou et al., 2017).The aforementioned distribution of the
cloud updraft velocity in the marine boundary layer of Finokalia, is consistent with values
observed in marine boundary layers (e.g. Albrecht et al., 1998 and references therein;
Fountoukis et al., 2007; Ghan et al., 2011), where they display a spectral dispersion around
zero value ($\sigma_w$ is calculated to be between 0.2 and 0.3 m s$^{-1}$).Thus, we can use the
characteristic cloud updraft velocity approach of Morales and Nenes (2010) when applying
the droplet parameterization to obtain the cloud updraft velocity PDF-averaged values of
cloud droplet number concentration and $s_{max}$. Moreover, a sensitivity test also considers a
more turbulent boundary layer ($\sigma_w = 0.6$ m s$^{-1}$), following Kalkavouras et al. (2017).
Furthermore, we determine the relative contribution of aerosol chemical composition, $\varepsilon\kappa$, and
aerosol number concentration, $\varepsilon N_{total}$, to variations in droplet number using a propagation of
variance (Sullivan et al., 2016; Bougiatioti et al., 2016; 2017),

$$\varepsilon N_{total} = \frac{\left(\overline{\frac{\partial N_d}{\partial N_{total}}}\sigma N_{total}\right)^2}{\sigma^2 N_d} \quad (6) \quad \text{and} \quad \varepsilon\kappa = \frac{\left(\overline{\frac{\partial N_d}{\partial \kappa}}\sigma\kappa\right)^2}{\sigma^2 N_d} \quad (7)$$

where $\sigma^2 N_d = (\overline{\frac{\partial N_d}{\partial N_{total}}}\sigma N_{total})^2 + (\overline{\frac{\partial N_d}{\partial \kappa}}\sigma\kappa)^2$ is the variance of the droplet number, $\sigma N_{total}$ is the standard deviation of the total aerosol number, $\sigma\kappa$ is the standard deviation of the hygroscopicity parameter, and $\overline{\frac{\partial N_d}{\partial N_{total}}}, \overline{\frac{\partial N_d}{\partial \kappa}}$ represent the average sensitivity of $N_d$ to aerosol number and hygroscopicity, respectively throughout a NPF episode, as calculated by the

droplet parameterization (Bougiatioti et al., 2016; 2017). The relative contribution of $\kappa$, and $N_{total}$ to the $N_d$ droplet number variation is estimated only during periods with high temporal resolution in chemical composition in order to capture the diurnal variability of $\kappa$ (ACSM measurements, May 2012 to May 2015).

## 2.6 Back-trajectories and meteorological data

For the entire dataset, three-dimensional back-trajectories have been calculated to determine the origin and trajectories of air-masses arriving at Finokalia. The HYSPLIT4 (Hybrid Single-Particle Lagrangian Integrated Trajectory; http://ready.arl.noaa.gov/HYSPLIT.php) back-trajectory model (Stein et al., 2015)was used. The back-trajectories initialized with meteorological conditions from GDAS (0.5° resolution), were calculated at several heights

(100, 500, and 1000 m above ground level (a.g.l.)), with a duration of 48 hours. The back-trajectories are important for understanding the provenance of the different air masses and how they related to the occurrence and evolution of NPF events. Meteorological parameters, as wind speed and direction, temperature, relative humidity, and solar radiation were also continuously monitored during the study period, by the automatic weather station installed at

Finokalia at 2 m a.g.l., and the time resolution for all of the measurements was 5 minutes (http://finokalia.chemistry.uoc.gr/).

## 3. Results and discussion

### 3.1 Aerosol chemical composition and hygroscopicity during NPF events

162 NPF episodes were recognized (Kalivitis et al., 2019) and the chemical composition of

submicron particulate matter during these episodes was primarily composed of sulfate, contributing on average by 39±8% to the total estimated $PM_1$ mass from June 2008 to December 2011 as derived from the respective bulk $PM_{10}$ 24-h quartz fiber filters, and by 51±12% from May 2012 to May 2015 as derived from the ACSM high-resolution measurements, respectively. Regarding the organic material its contribution to the total

estimated $PM_1$ mass was found to be in the order of 38±10% using the bulk $PM_{10}$ 24-h quartz fiber filters, and to the total $PM_1$ mass was calculated to be 44±12% using the ACSM data, indicating that the relative abundance of sulfate and organics dictate to a high extent the

hygroscopic and cloud-activating properties of submicron particles over Finokalia. Figure S3 shows that sulfate contributed to a greater fraction of the aerosol during summer and autumn, and to an almost equal extent in winter and springtime when considering the daily 24-h quartz fiber filters and the ACSM data, respectively. On the contrary, for both chemical composition techniques, organic material contributed more during winter due to the long-range transport of organic-rich material from the Greek mainland, whilst its contribution was minimum during autumn.

Following Section 2.4, $\kappa$ was calculated using the chemical composition data. The predicted $\kappa$ derived from the estimated PM$_1$ varied from 0.21 to 0.52, with a mean value of 0.38±0.06, while when the ACSM data were considered, $\kappa$ varied from 0.20 to 0.45, with a mean value of 0.36±0.06. This insignificant difference regarding the $\kappa$ is due to the lower values of organic and inorganic volume fractions $\varepsilon_{org}$, $\varepsilon_{inorg}$ derived from the ACSM data. Mean $\kappa$ values were estimated to be somehow lower in winter and higher during autumn, while in spring and summer the average aerosol hygroscopicity exhibited generally similar values. Indicatively, the diurnal variability of the $\kappa$ derived from the chemical composition analysis and from the CCN data for supersaturations below 0.20%, and for supersaturations ranging from 0.20 to 0.40, 0.40 to 0.50, and 0.60 to 0.70% on 29 August 2012 is presented in Figure S4. It can be seen that $\kappa$ exhibited lower values throughout the morning hours (06:00 to 09:00 LT), and tended to increase between 12:00 and 21:00 LT, when considering the data derived from the ACSM, and the CCN counter for each critical supersaturation. This increase regarding the $\kappa$ can be ascribed to  the downward transport of secondary organic aerosol (SOA) during the boundary layer mixing, whilst at some point after noon, $\kappa$ begun to augment probably linked to the formation of particulate sulfate during this period. In particular, the increase was estimated to be as 21% when the ACSM data were considered, and 21, 24, 29, 69 and 42% for supersaturations under 0.20%, from 0.20 to 0.40%, from 0.40 to 0.50%, and from 0.60 to 0.73%, respectively when the CCN data were used. As expected, lower supersaturation levels are associated with higher $\kappa$ values, indicating that smaller particles were much less hygroscopic than larger ones, with an average difference being of 0.2-$\kappa$ units between the lower (under 0.20%) and the maximum supersaturation (0.60-0.70%). This feature has been attributed to the enrichment of organic material in sub-100 nm particles (Kalivitis et al., 2015). The chemically-derived $\kappa$ from the ACSM measurements generally does not present any remarkable fluctuation (see grey crosses in Fig. S4), and it seems to converge better with the CCN-derived $\kappa$ values for supersaturations varying from 0.20 to 0.40%, compared to the other supersaturations. This relative constant character of the chemically-derived $\kappa$, may be an evidence that using prescribed levels of supersaturation or critical diameters to calculate CCN number concentrations can provide a biased influence of NPF events on CCN, since the pre-

assigned $s$ or $d_c$ are essentially different from those occurring in the "real" cloud-forming
conditions (see below the Section 3.3).

## 3.2 Characteristics and interpretation of the Finokalia NPF events

In several studies to date (summarized in the introduction), NPF impacts on CCN number
concentrations is based on analysis of the evolution of the aerosol size distribution over time,
to quantify $i$) how long it takes before freshly-formed particles in a given air-mass reach
CCN-relevant sizes, and, $ii$) the degree to which CCN number concentrations are augmented
from the NPF. Here we present in detail the corresponding methodology used to interpret the
NPF data from Finokalia, by applying to a "representative" type-I NPF event (according to
the Dal Maso et al., 2005 classification) observed at Finokalia on 29 August 2012 (Fig. 1),
where the subsequent growth of the aerosols generates a characteristic "banana shape" in the
time-series of diurnal particle number concentration (Fig. 1a). The episode was characterized
by a burst in particle number concentration in the 9 to 25 nm diameter range (nucleation-
mode), and enables a robust determination of the starting time ($t_{start}$) of the NPF event. Since
we had no means to determine the intermediate negative-ion concentrations we modified the
concept of Leino et al. (2016) using the intermediate nucleation-mode particles, which
corresponds to particles with diameters from 9 to 25 nm in order to determine the initiation of
a NPF event. We calculated half-hour median concentrations of the nucleation-mode particles
from the measurement data, since the half-hour median concentration was deemed sufficient
to determine the $t_{start}$. When plotting the time series of the intermediate nucleation-mode
particles, the NPF is distinctly visible as the particle concentrations rapidly increase from
3,850 to just over 17,000 cm$^{-3}$ over a 2.5 h period starting at 08:30LT (Fig. 1b). The starting
of the NPF is further confirmed by the evolution of the particle size distribution ("banana
shape" pattern; see Fig. 1a) when the new 9–nm particles appear and shift gradually towards
to larger sizes. The nucleation-mode particles peak at 11:00 LT (see Fig. 1c), without any
visible change in the Aitken-mode concentrations until after 11:30 LT. This increase, in
conjunction with the decrease of the nucleation-mode particles in number, strongly suggests
the transfer of nucleation-mode to Aitken-mode particles owing to condensation and
coagulation. The NPF event is said to terminate when the nucleation-mode particles start to
decrease. The appearance and formation of the nucleation-mode particles are linked to the
onset of solar radiation (Fig. 2). Afterwards, particles continued to grow faster in size for
several hours, consistently with finding in other studies (e.g. Paasonen et al., 2018), exceeding
100 nm in diameter at 21:30 LT. Following the methodology of the mode-fitting (Hussein et
al., 2004; Kulmala et al., 2012) the nucleation-mode particles exhibited a growth rate of 3.7
nm h$^{-1}$, while the formation rate value of particles in the nucleation-mode was 2.0 cm$^{-3}$ s$^{-1}$

(Kulmala et al., 2012), which are well in the range of the representative values reported by
Kalivitis et al. (2019) at Finokalia site.

To quantify the impact of NPF on CCN number concentrations, the following approach is used. From the time-series of the aerosol size distribution and chemical composition that spans each NPF event, the time-series of CCN concentration for a number of supersaturations $s$, $CCN_s$, is calculated following Section 2.4. It should be noted that, from June 2008 to December 2011 when daily bulk $PM_{10}$ quartz fiber filters were used, there was only one $\kappa$ available for each NPF day. We then determine the starting time, $t_{start}$, and its corresponding CCN concentration, $CCN_{s, t_{start}}$. The enhancement of CCN from the NPF at supersaturation $s$, $R_s$, is then calculated by normalizing the CCN time series with $CCN_{s, t_{start}}$ for each NPF event, $R_s = \frac{CCN_s}{CCN_{s,t_{start}}}$. By definition, the $R_s$ is equal to unity at $t_{start}$ and theoretically should remain so until the "wave" of new particles reaches a large enough size to influence $CCN_s$.

Figure 3a, presents the evolution of the $R_s$ for each supersaturation against aerosol number concentrations, before, during and after the event. From 08:30 LT ($t_{start}$) and for 5 hours later (13:30 LT), the $R_s$ displays a similar pattern, especially for supersaturations above 0.38%, with values ranging from 0.75 to 1.32 (average 1.00±0.06) according to the conceptual model. This pattern reveals that during the morning hours and until 13:30 LT, the estimated CCN number concentrations exhibit almost equal values for each supersaturation, since the denominator is constantly the same. At 13:30 LT, the $R_s$ acquires different values in a given supersaturation as depicted in Figure 3a. This time is crucial in order to estimate the initiation of the influence on the potential CCN due to NPF, and is termed the "decoupling time", $t_{dec}$. We determine the $t_{dec}$, and therefore the period (i.e. start and end) of intense NPF impact on the CCN spectrum, based on the temporal evolution of the relative dispersion (RD) of the $R_s$ for all supersaturations (Fig. 3b). RD was calculated by dividing the standard deviation of the instantaneous values of the $R_s$ (at 0.10, 0.38, 0.52, 0.66, 0.73, and 1.00% supersaturation) with their average value at each time step (5-min temporal resolution). RD is useful, at it is highly sensitive to the introduction and evolution of particles from NPF as they transit the distribution over the resolved supersaturation range. It is said that, NPF influences the CCN as long as the RD exceeds the envelope of (low) values seen during the initial stages of the NPF event. Indeed, from 08:30 to 13:30 LT, the RD is low (generally less than 0.1), and rapidly increases at 13:30 LT and on – indicative of the large spread in $R_s$ from the influence of NPF on the production of particles which activate at larger supersaturations; therefore 13:30 LT corresponds to the $t_{dec}$. The impact of NPF on the CCN spectrum is terminated when the RD drops to values seen prior to $t_{dec}$ (21:30 LT, see Fig. 3b), presumably when the NPF has evenly affects CCN number concentrations at all $s$ levels. However, it should be clarified that this "end time" (e.g. 21:30 LT) is identified on the day of the NPF episode, since we had no

real means to record the continued growth processes due to NPF from a previous day and beyond the point of the influence of NPF on the droplet's formation ($t_{Nd}$, see in Section 3.3). The elevated RD seen after 23:00 LT may be a result of residual NPF particles mixing in the air-masses sampled at Finokalia, or a result of other small-scale variations (from local sources) in the CCN spectrum.

Subsequently, we calculate the evolution of the $R_s$ before and after the $t_{dec}$ for each supersaturation on 29 August 2012 (Fig. 3a). Specifically, "before" is the time period between $t_{start}$ and $t_{dec}$, whereas "after" is the period from the $t_{dec}$ until the end of CCN production (21:30 LT). This variation of the $R_s$ indicates, for each supersaturation value, the increase of the CCN number concentration due to particles originating from the NPF. The $R_s$ was estimated to be

0.89±0.09, 0.94±0.08, 1.02±0.09, 1.04±0.09, 1.03±0.09, and 0.99±0.08 prior to the starting of the CCN production (i.e. between 08:30 and 13:30 LT), and 0.90±0.23, 1.09±0.60, 1.21±0.52, 1.25±0.43, 1.26±0.40, and 1.39±0.32 for 0.10, 0.38, 0.52, 0.66, 0.73, and 1.00% supersaturation, respectively after 13:30 LT and until the end of the production. The time intervals and $t_{dec}$ are driven by the processes that affect the aerosol number distributions (i.e.

coagulation and condensation), and hence affect the CCN population. Assuming a constant growth rate (e.g. 3.7 nm h$^{-1}$), we estimate the duration time after the $t_{start}$, during which the freshly formed particles need to grow in size reaching the respective $d_c$ (35, 43, 46, 54, 67, and 162 nm for $s$ 1.00, 0.73, 0.66, 0.52, 0.38, and 0.10%. respectively)and act as CCN. Considering an initial diameter of 9 nm for the newly formed particles at $t_{start}$, $t_{dec}$ appears 7 to

41 h after the $t_{start}$ for supersaturations between 1.00 and 0.10%, and from 2.7 to 37 h when an initial diameter of 25 nm is considered for the same supersaturation range. This feature shows that, when only constant growth rate is considered, the freshly nucleated atmospheric particles attain the largest sizes ($d_c$=162 nm when $s$=0.10%) during the night (21:30 LT) of the following day (30$^{th}$ August), and early in the morning (01:30 LT) on the 31$^{st}$ August, when 25

and 9 nm were considered as initial diameters at $t_{start}$, respectively. Therefore, it is apparent that it takes longer time compared to the RD methodology to observe the influence of NPF on the concentration of particles which are able to act as CCN at lower supersaturations (e.g. 0.10%), when the GR is the sole factor determining the time delay from $t_{start}$ to $t_{dec}$. Observed $t_{dec}$ is generally earlier compared to the aforementioned values, and this temporal

inconsistency may occur owing to the previous consideration of a constant growth rate, since the growth rate has the ability to change, and specifically to increase with an increasing particle diameter (Paasonen et al., 2018). Concurrently there are also several microphysical processes (i.e. the synoptic wind flow, the boundary layer dynamics, the presence of pre-existing particles) which influence the time lag between $t_{start}$ and $t_{dec}$.

The $R_s$ exhibits almost similar mean values after the $t_{dec}$ until 21:30 LT, for 0.52, 0.66, and 0.73% supersaturation. Thus, the number of the newly-formed particles which reach the CCN-size ($d_c$ varying from 43 to 54 nm) is independent from $s$, indicative of the assumption of a similar chemical composition for all sizes, or could be merely that particle number in the size range between 43-54 nm particles increased more or less to the same extent, after the $t_{dec}$.

Using the above-mentioned values of the $R_s$, we determined the subsequent percentage increase of the CCN number concentrations related to particles originating from NPF. The enhancement of the CCN number concentrations was calculated to be 1, 16, 19, 20, 22, and 40% for $s$ 0.10, 0.38, 0.52, 0.66, 0.73, and 1.00%, respectively. For supersaturation of 0.10% the increase was merely 1%, while for the supersaturations 0.38, 0.52, 0.66, and 0.73 the

augmentation was generally the same, which is consistent with the similar $R_s$ observed in the same size range, as mentioned above .Regarding the $s$ of 1.00%, the aerosol sizes are even smaller (~35 nm) and the contribution of NPF on CCN increases considerably. When looking at the diurnal evolution of the aerosol size distribution (Fig. 1a), particles in the size range of around 35 nm also pre-existed the NPF event ($t_{start}$) and could contribute to CCN number

concentrations. These contributions are suggestive of the convolution of NPF with condensational growth of both fresh and pre-existing ("background") particles , introducing an upper limit of bias of approximately  50%, which could originate from the pre-existence of large enough particles (not originating from NPF) that can grow to CCN-relevant sizes. The amount of the "background" particles, which are large enough and also have sufficient time to

grow to CCN, was calculated by subtracting the mean value of the concentration of particles in the nucleation-mode from $t_{start}$ until 11:30 LT (the formation of the nucleation-mode particles ceased - Fig. 1a) and the respective mean value2 hours prior to the $t_{start}$. The latter depicts that the impact of NPF on CCN number concentrations, and subsequently on cloud properties, also depends on the background conditions (clean vs polluted air). Under clean air

conditions (limited pre-existing particles preceding the NPF), which constitute the 40% of the NPF days, it has been found that CCN concentrations are enhanced by 45 to 80% in the 0.10-1.00% supersaturation range, compared to more polluted conditions.

The procedure outline in Section 3.2 is repeated for the all the 161 remaining NPF episodes to determine the increase of the CCN number concentrations owing to particles originating from

the NPF episodes. The comprehensive results are presented in Table S2, and an extensive seasonal analysis in the Supplementary Material 3.2 (SM 3.2). Altogether, when considering all 162 NPF episodes we found that, the average contribution of NPF to the CCN budget over the eastern Mediterranean varied from 29 to 77% in the 0.10-1.00% supersaturation range, and displayed a seasonal variation (Fig. 4). In winter, $t_{start}$ was observed during daytime

(median 11:00 LT), followed by $t_{dec}$ 2.5 hours later. The augmentation on CCN number

concentrations due to the atmospheric NPF and growth was estimated to be 30, 43, 43, 44, 46, and 54% for 0.10, 0.38, 0.52, 0.66, 0.73 and 1.00% supersaturation, respectively (Fig. 4). For spring and summer, $t_{start}$ exhibited a median value at 10:00 LT, and 09:00 LT, respectively, whilst the $t_{dec}$ was on average 2.5 hours after the $t_{start}$. The CCN production associated with the

nuclei growth to larger sizes increase by almost 41% for both seasons (Fig. 4), and for the aforementioned supersaturations. Finally, throughout autumn, $t_{start}$ was detected in the morning (median 09:30 LT), followed by $t_{dec}$ on average 3.5 hours after the $t_{start}$, whereas the NPF episodes elevated the CCN numbers by 29, 47, 52, 55, 58, and 77% (Fig. 4) for each supersaturation, respectively. Hence, according to the above conceptual model, NPF taking

place in the eastern Mediterranean may considerably influence CCN numbers (compared to levels prior to $t_{dec}$), for prescribed levels of superasaturation. According to Kalivitis et al. (2019), higher growth rates are calculated for summer and autumn, compared to winter and spring. Consequently, it would be expected that the time delay between $t_{start}$ and $t_{dec}$ would be lower during summer and autumn, if only the influence of growth rate is taken into account.

Nevertheless, the GR is not entirely responsible for the growth of the freshly nucleated atmospheric particles into CCN-relevant sizes and cloud droplets, and further microphysical processes favor the NPF and consequently determine the $t_{dec}$, as we have seen above. The air-masses reaching at Finokalia during summer, contain significant amount of pre-existing particles (before $t_{start}$ on average higher by 58% compared to winter and spring, with larger

load in the Aitken-mode) providing a sink for newly-formed particles via condensation and coagulation (Dameto de España et al., 2017), a feature which has an impact on the growth of the freshly-formed particles to larger sizes, and therefore also determines $t_{dec}$ as already seen in the RD analysis.

## 3.3 Impact of NPF on droplet number and cloud formation

Following the proposed methodology (Section 2.5), we estimated the number of droplets ($N_d$) and the maximum supersaturation ($s_{max}$) that would form in a cloud, based on the aerosol number size distribution ($N_{total}$), chemical composition ($\kappa$), and cloud updraft velocity ($\sigma_w$) throughout each NPF event. Results of $N_d$ are shown in Figure 5 for cloud updraft velocities of 0.3 m s$^{-1}$ (bottom) and of 0.6 m s$^{-1}$ (top), while Figure 6 depicts the corresponding $s_{max}$

during the "representative" NPF event recorded at Finokalia on 29 August 2012. As expected, the higher cloud updraft velocity generates larger values of both $s_{max}$, and $N_d$. On the time period between 08:30-17:25 LT, which includes the formation (08:30-11:00 LT) and growth hours (after 11:00 LT) of the episode, as well as the starting of the CCN influence due to NPF (13:30 LT), the arrival of the air-mass is followed by a depression in $N_d$ (relative mean

decrease 7.9±2.9% for $\sigma_w$=0.3 m s$^{-1}$ and 13.5±3.9% for $\sigma_w$= 0.6 m s$^{-1}$). Concurrently, there is a slight increase in the maximum supersaturation (relative mean increases 4.7±2.1% for $\sigma_w$=0.3

m s$^{-1}$ and 6.9±2.3% for $\sigma_w$= 0.6 m s$^{-1}$). Both trends are related to decreases in the accumulation-mode aerosol number (from 08:30 to 17:25 LT, see Fig. 1c, right-hand y axis), owing to processes other than NPF (e.g. development of the boundary layer, dry deposition) –

as the latter has not had the chance to influence particles that act as CCN in clouds. For both cloud updraft velocities the $s_{max}$ was calculated to be under 0.17% highlighting the low levels of $s_{max}$ developed. Hence, according to these low values of supersaturation formed in the clouds, and in conjunction with the mean $d_c$ at 162 nm for 0.10% supersaturation, it is clear that most of the activated droplets belong to the accumulation-mode particles. $N_d$ exhibits the

minimum value at 17:25 LT (Fig. 5), coinciding with when $s_{max}$ begins decreasing (CCN start to grow further to form droplets, and they compete for water vapor thus decreasing $s_{max}$), depecting the moment when droplet formation begins to "feel" the particles generated from NPF. Furthermore, this time stamp also coincides with the time when the number concentration of particles in the accumulation-mode exhibits the lowest value as well (see Fig.

1c). Hereafter, this time will be expressed as $t_{Nd}$ (Fig. 5). There is a time lag between $t_{dec}$ and $t_{Nd}$, since particles formed in a NPF event need sufficient time to grow into CCN-relevant sizes, and subsequently into a cloud droplet. After $t_{Nd}$, $s_{max}$ is negatively correlated with $N_d$ for both cloud updraft velocities, due to the increasing competition for water vapor from the growing number of CCN. For both cloud updraft velocities, the increase of $N_d$ until the

midnight  was similar and on the order of  20.0±6.5%, leading to a simultaneous decrease of $s_{max}$ by 6.0±2.7% (Table S3). Interestingly, water vapor competition effects can be assessed by comparing the number concentration of $N_d$ with the estimated CCN number for a supersaturation equal to the value of $s_{max}$ at the time of $t_{Nd}$ (where competition effects from the NPF-generated particles are vanishingly small) for the time period between $t_{Nd}$ and midnight.

Using this approach, and by comparing the derived estimated CCN with the respective $N_d$ (assuming that without competition effects all CCN would activate in droplets), we find that competition effects suppress $N_d$ by 20% for $\sigma_w$=0.3 m s$^{-1}$ and 12.3% for $\sigma_w$=0.6 m s$^{-1}$. It is worth noting that, if $s_{max}$ did not vary over the period of $N_d$ influence, the increase of $N_d$ from the $t_{Nd}$ until midnight was similar for both $\sigma_w$ and merely of 5.5±2.5%, since the competition

for water vapor is restricted considerably. The above clearly shows that the prescription of a constant supersaturation in the CCN analysis may lead to biased results regarding the impact of NPF on regional clouds. Since $N_d$ does not increase significantly but until midnight, it is clear that most of the impact of the NPF is on nocturnal clouds, which carries important implications for the formation of drizzle and structure of the boundary layer in the following

day.

The degree to which $N_{total}$ and $\kappa$ variations influences $N_d$ variability can be expressed by calculating the relative contribution of the total aerosol number, and the hygroscopicity to the

droplet number using the equations (6),, and (7) in Section 2.5. The results are displayed thoroughly in Table S4. We find that the variance of the droplet number (see in Section 2.5) from $t_{Nd}$ to midnight exhibits an average value of 30 cm$^{-3}$ when $\sigma_w$ is equal to 0.3 m s$^{-1}$, and 35 cm$^{-3}$ for $\sigma_w$ equal to 0.6 m s$^{-1}$, respectively. 68% of this variance can be attributed to aerosol number and the remaining 32% to changes the chemical composition. The above procedure, when carried out for the 161 remaining NPF episodes, provides consistently similar results (Results depicted in Table S3) for both cloud updraft velocities examined. A detailed summary of the analysis by episode and season is presented in the Supplementary Material 3.3 (SM 3.3).

Overall, during the 162 NPF days, the $s_{max}$ formed in clouds augments slowly after the $t_{start}$ and decreases gradually - after the $t_{Nd}$ - when the particles from NPF begin contributing to $N_d$. After the $t_{Nd}$, the mean value of the $s_{max}$ was calculated to be 0.11±0.03%, and 0.15±0.05% for cloud updraft velocities ($\sigma_w$) of 0.3 m s$^{-1}$, and 0.6 m s$^{-1}$, respectively. Concurrently, $N_d$. is influenced from the afternoon and on, and their average increase due to the NPF varied from 1 to 55%, and from 0.2 to 62% for each $\sigma_w$, respectively (Table S3). In wintertime, $t_{Nd}$ was observed in the afternoon (median value at 17:30 LT). A slight decrease of the $s_{max}$ was calculated after the $t_{Nd}$, compared to the period between $t_{start}$ and $t_{dec}$ (10% for $\sigma_w$=0.3 m s$^{-1}$ and 9% for $\sigma_w$=0.6 m s$^{-1}$), whilst the respective increase regarding the $N_d$ due to the NPF episodes was estimated to be 13% and 17%, for the aforementioned $\sigma_w$ (see Table S5). For spring, the $t_{Nd}$ showed a median value at 15:40 LT. $s_{max}$ decreases by 10% and 7.5% for both cloud updraft velocities, whilst the expected augmentation of the $N_d$ compared to pre-$t_{Nd}$ values during the NPF days was calculated to be 12% (for $\sigma_w$=0.3 m s$^{-1}$) and 15% (for $\sigma_w$=0.6 m s$^{-1}$). Throughout summer, $t_{Nd}$ occurred at 15:00 LT (median value). For both $\sigma_w$ the decrease of $s_{max}$ caused by the NPF was on average all the same (10%), whereas at the same time the NPF is followed by a limited augmentation (7% and 9%, for $\sigma_w$ equal to 0.3 m s$^{-1}$ and 0.6 m s$^{-1}$, respectively) regarding the $N_d$. In autumn, the $t_{Nd}$ displayed a median value at 16:30 LT, and the variations regarding the $s_{max}$ and $N_d$ are similar with the respective values calculated during spring (see Table S5). Lastly, from the relative contribution of the total aerosol number and chemical composition to $N_d$, it can be seen that in all seasons, the variance of the total aerosol number (on average 91%) dominates, with chemical composition contributing the remaining 9% droplet number variance (Table S5).

## 4. Summary and Conclusions

The aerosol particle number size distributions along with chemical composition and meteorological parameters were studied at a remote background site in the eastern Mediterranean over a 7-year period in order to quantify how regional new particle formation

(NPF) events modulate the concentration of aerosol, cloud condensation nuclei (CCN), droplet number and maximum supersaturation developed in clouds of the region.

Overall, 162 NPF episodes were recorded with the majority occurring during spring and summer (32 and 30.8%, respectively), few during winter (14.8%) and the rest (22.4%) during autumn. The timing and duration of NPF influences on the CCN spectrum and cloud droplet number concentration were determined using a set of new statistical metrics derived from the observational data. Wintertime NPF events were found to start around 11:00 LT ($t_{start}$) and

begin to increase the CCN number concentrations 3 hours after the $t_{start}$, while in springtime were initiated one hour earlier and start to increase the CCN number concentrations 2.5 hours after the $t_{start}$. During summer, the recorded NPF events started the earliest (09:30 LT) and the augmentation on the CCN number concentrations occurred roughly 2.5 h after the $t_{start}$, while in autumn NPF episodes were observed between 09:30 and 10:00 LT, but with the largest

delays regarding the increase of the CCN number concentrations - 3h 30 min after the $t_{start}$. Generally, when accounting for all NPF episodes, we found that the average increase on CCN levels (0.10-1.00% supersaturation) from the NPF over eastern Mediterranean ranged from 29 to 77%, with air-masses containing lower amount of pre-existed particles (cleaner air) exhibiting a higher increase in the CCN number concentration, and consequently to the cloud

droplet number concentrations due to the NPF

When the observed size distributions and chemical composition are used in conjunction with a cloud droplet parameterization, the impact of NPF on $N_d$ differs considerably from the CCN-based analysis. Regardless of season, we find that the maximum supersaturation developed in typical boundary layer clouds in the eastern Mediterranean (cloud updraft

velocities of the order of 0.3 m s$^{-1}$) vary between 0.07% and 0.12%, giving on average cloud droplet number increases of 7% to 13%. This 4 to10-fold decrease in $N_d$ sensitivity to NPF (compared to what is deduced from the CCN analysis) is primary from the actual cloud supersaturation being much lower than the prescribed levels in the CCN analysis. $N_d$ sensitivity to NPF however is further reduced during the evolution of NPF events owing to

their increased competition for water vapor when forming cloud droplets (the droplet response can be suppressed by almost 1/5 compared to assuming constant supersaturation throughout the NPF). Nevertheless, most of this droplet variability is driven by changes in aerosol number (91%), the rest being driven by composition changes. The lowest impact on $N_d$ is observed during summer, as this season exhibits the highest aerosol concentrations prior to

NPF events - that either act as CCN or grow to become so during an event. Pre-existing particles have been estimated to contribute up to 50% of the activated CCN during summer, denoting the importance of background conditions. A striking consequence of the low cloud supersaturations is that NPF impacts on $N_d$ are observed much later in the event, typically in

the afternoon (after 15:00 LT), and that $N_d$ is relatively insensitive to increases in CCN during the course of an event owing to the competition effects for water vapor. Thus, the impacts of NPF events on eastern Mediterranean clouds occur during the late evening and nighttime. Although such $N_d$ enhancements may limit the short-term impact of NPF on shortwave cloud forcing – it may reduce cloud drizzle and promote stabilization of the marine boundary layer with potentially important implications for the overall boundary layer structure (e.g. Rosenfeld et al., 2006) in days following NPF events.

Perhaps one of the most important findings of this study is the importance of constraining the levels of supersaturation that are generated in ambient clouds, and the diurnal characteristics of the influence of NPF events on cloud properties. Even though the events themselves can occur early in the day, CCN number concentrations start becoming affected after 2-3 hours and CDNC much later, in the late afternoon and early evening. Thus, choosing prescribed levels of supersaturation or diameters to define CCN number concentrations can provide substantially biased or incomplete insights on the influence of NPF events on regional clouds, the hydrological cycle, and climate. The approach presented here offers a simple and effective paradigm for quantifying the potential impacts of NPF events on clouds, with tools available to interested researchers upon request.

**Author contributions**

AN and AB conceived the study and developed the analysis tools. AB, NK and NM contributed measurements. AN, AB, PK carried out the analysis and AN, AB, PK, NM wrote the paper. All authors commented on the manuscript.

**Acknowledgments**

This research is co-financed by Greece and the European Union (European Social Fund- ESF) through the Operational Programme «Human Resources Development, Education and Lifelong Learning» in the context of the project "Reinforcement of Postdoctoral Researchers" (MIS-5001552), implemented by the State Scholarships Foundation (IKY).This study also received financial support from the PANhellenic infrastructure for Atmospheric Composition and climatE change" (MIS 5021516) which is implemented under the Action "Reinforcement of the Research and Innovation Infrastructure", funded by the Operational Programme "Competitiveness, Entrepreneurship and Innovation" (NSRF 2014-2020) and co-financed by Greece and the European Union (European Regional Development Fund). We also acknowledge project PyroTRACH (ERC-2016-COG) funded from H2020-EU.1.1. - Excellent Science - European Research Council (ERC), project ID 726165

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

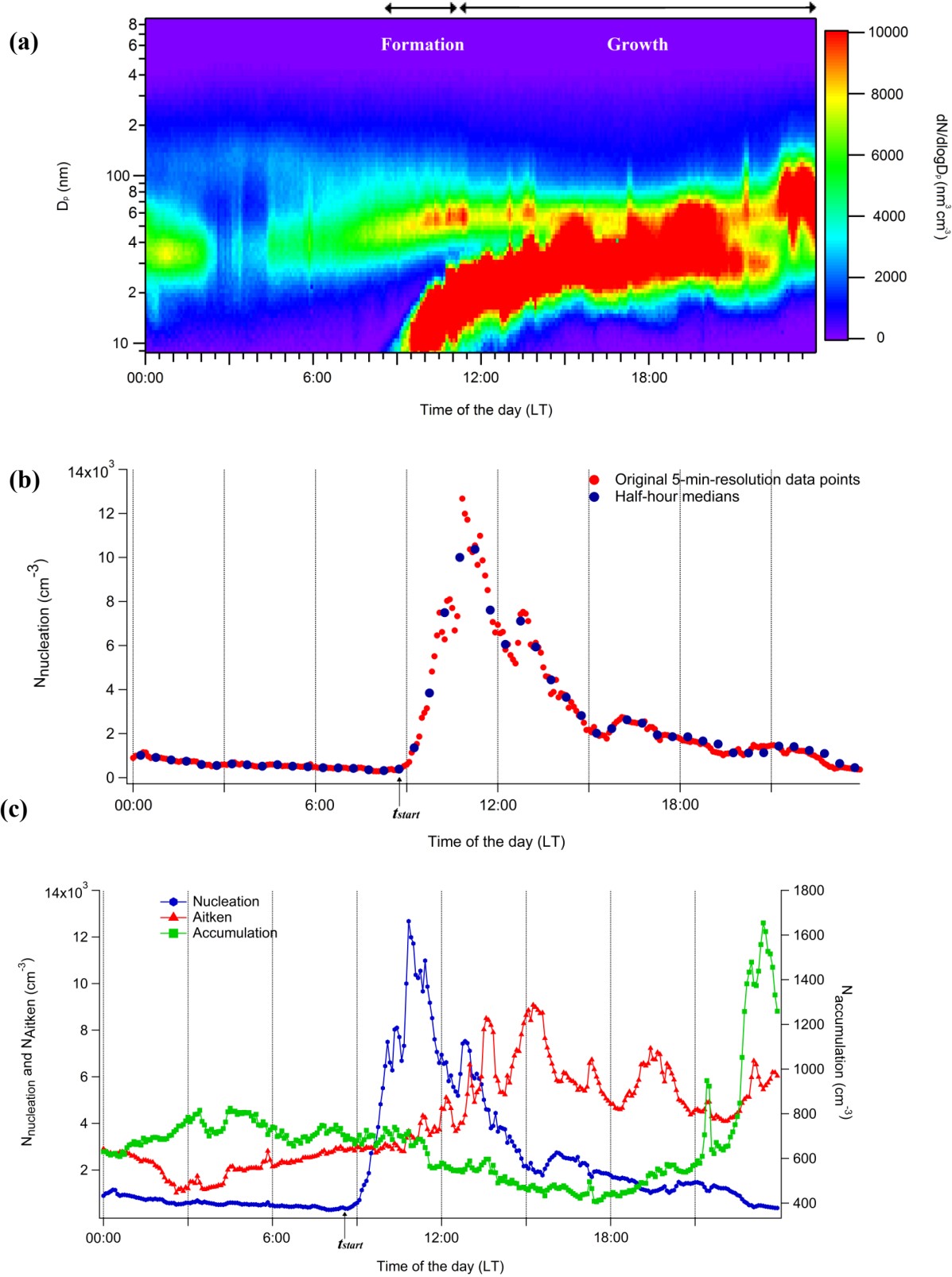

**Figure 1.** A "representative" new particle formation event captured at Finokalia on 29 August 2012. **(a)** Diurnal evolution of the aerosol size distribution, **(b)** temporal evolution between 5-min-resolution original points (red dots) and calculated half-hour median concentrations of particles in size range of 9-25 nm (blue dots), and **(c)** diurnal evolution of nucleation (blue line), Aitken (red line), and accumulation mode particle number concentration (green line)**,**respectively.

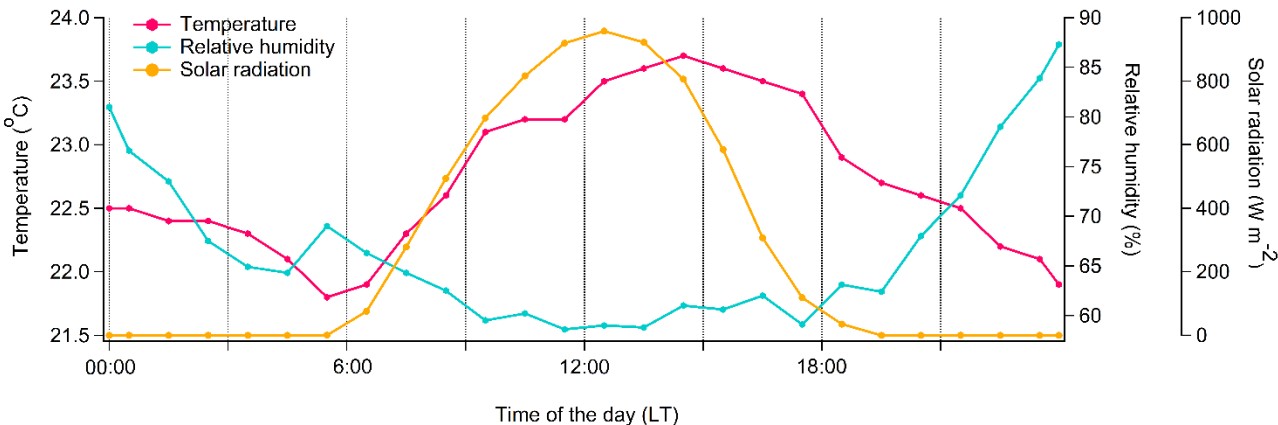

**Figure 2.** Diurnal evolution of the temperature (purple line), relative humidity (light blue), and solar radiation (orange line), respectively during the "representative" new particle formation event observed at Finokalia on 29 August 2012.

**(a)**

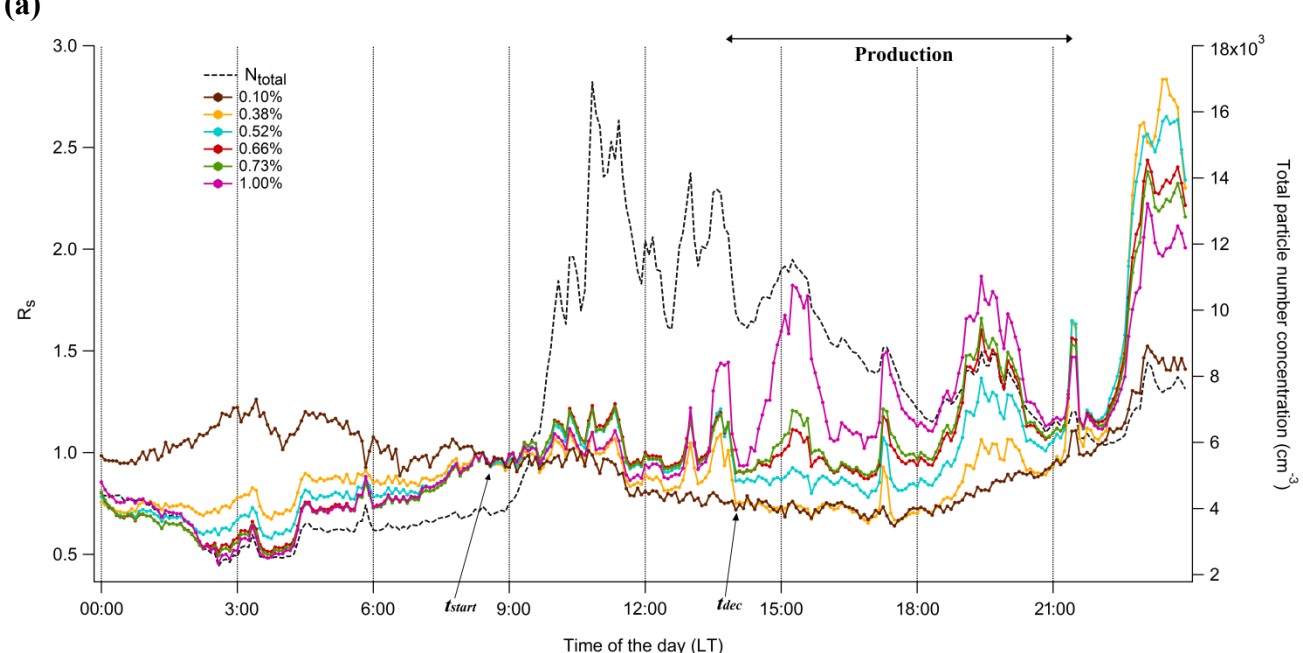

**(b)**

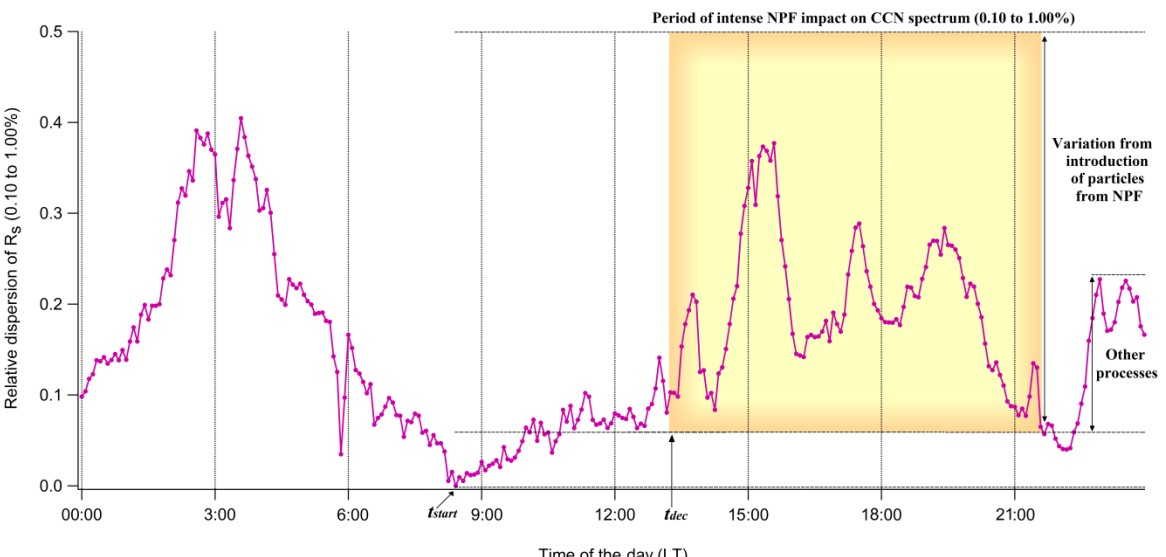

**Figure 3. (a)** Diurnal evolution of the $R_s$ for supersaturation 0.10 (brown line), 0.38 (orange line), 0.52 (light blue line), 0.66 (red line), 0.73 (green line), and 1.00% (purple line) (left axis) respectively, and total particle number concentrations, $N_{total}$ (black line-right axis) during the "representative" new particle formation event captured at Finokalia on 29 August 2012. **(b)** Diurnal evolution of the relative dispersion (RD) of the $R_s$ for all supersaturations ($s$) (0.10 to 1.00%) during the "representative" new particle formation event captured at Finokalia on 29 August 2012. $t_{start}$ (08:30 LT) is the starting time of the NPF event, while $t_{dec}$ is the "decoupling time" (13:30 LT), when the NPF episode start to influence the CCN concentrations according to the approach described in the main text. The period of intense NPF event on CCN spectrum for all $s$ is shaded in yellow.

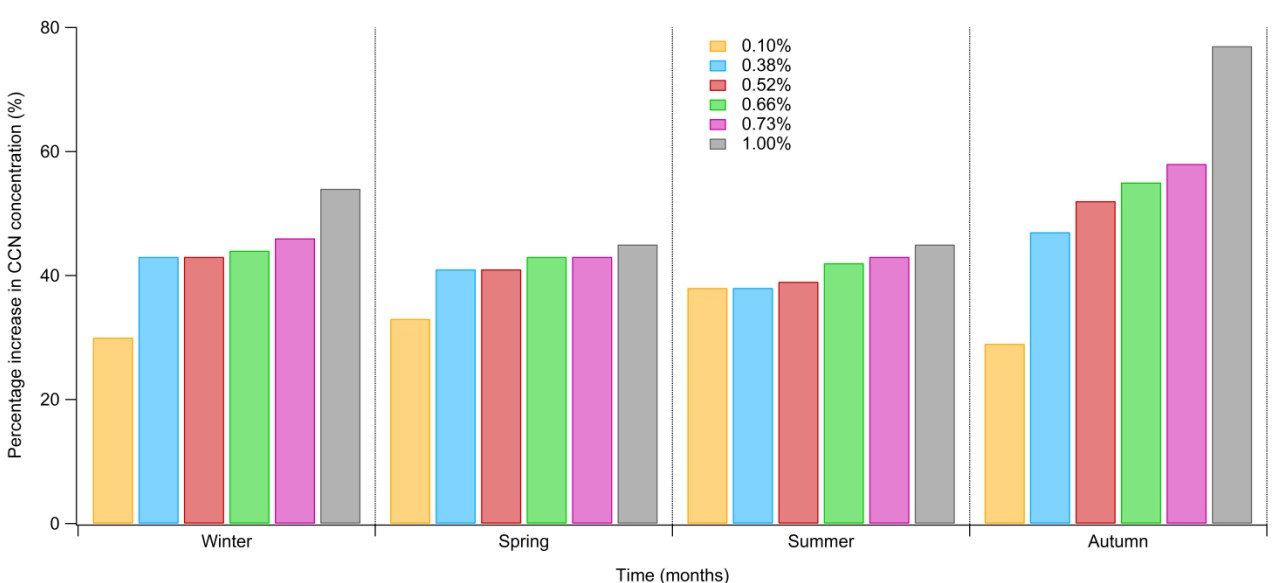

**Figure 4.** Seasonal variation of percentage increase regarding the estimated CCN concentrations for supersaturation 0.10 (orange bars), 0.38 (light blue bars), 0.52 (red bars), 0.66 (green bars), 0.73 (purple bars), and 1.00% (grey bars), respectively, relative to the available 162 NPF days at Finokalia, throughout the period June 2008-May 2015.

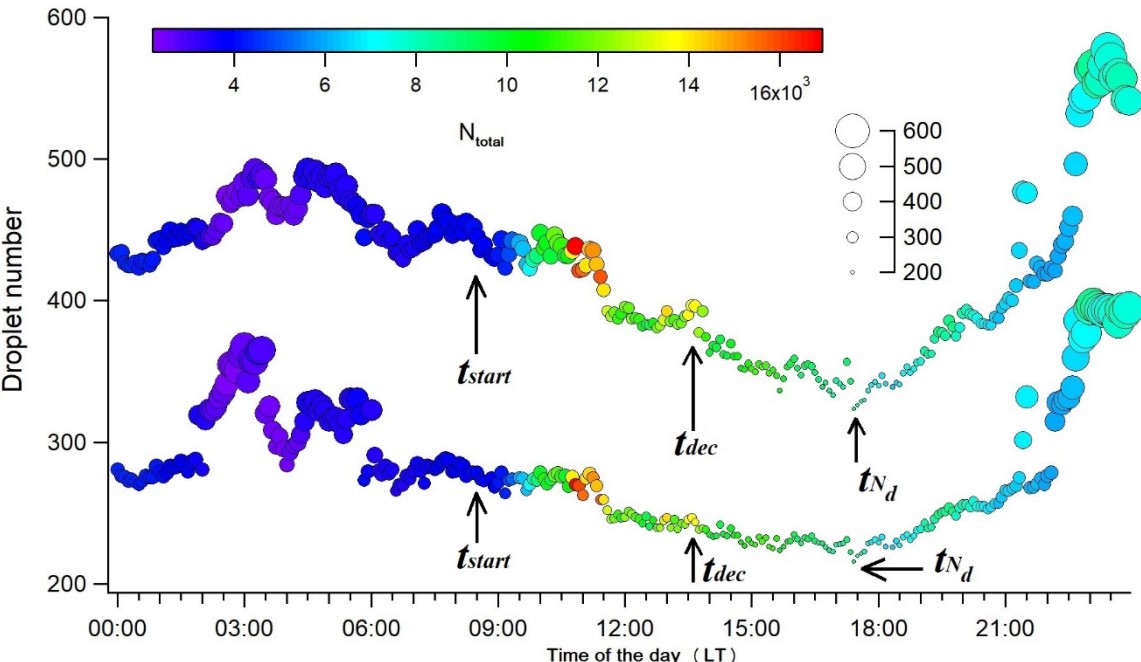

**Figure 5.** Diurnal evolution of the total aerosol particle number concentrations in $cm^{-3}$ ($N_{total}$– color bar) and calculated cloud droplet number concentrations ($N_d$) (left axis) for updraft velocities of $\sigma_w$= 0.3 m s$^{-1}$ (bottom), and $\sigma_w$=0.6 m s$^{-1}$ (top) during the "representative" new particle formation event captured at Finokalia on 29 August 2012. The size of the circles corresponds to the number concentration of $N_d$, while $t_{dec}$ is the "decoupling time" (13:30 LT), and $t_{Nd}$ is the time when the number of droplets start to "feel" the NPF (17:25 LT), according to the approach described in the main text.

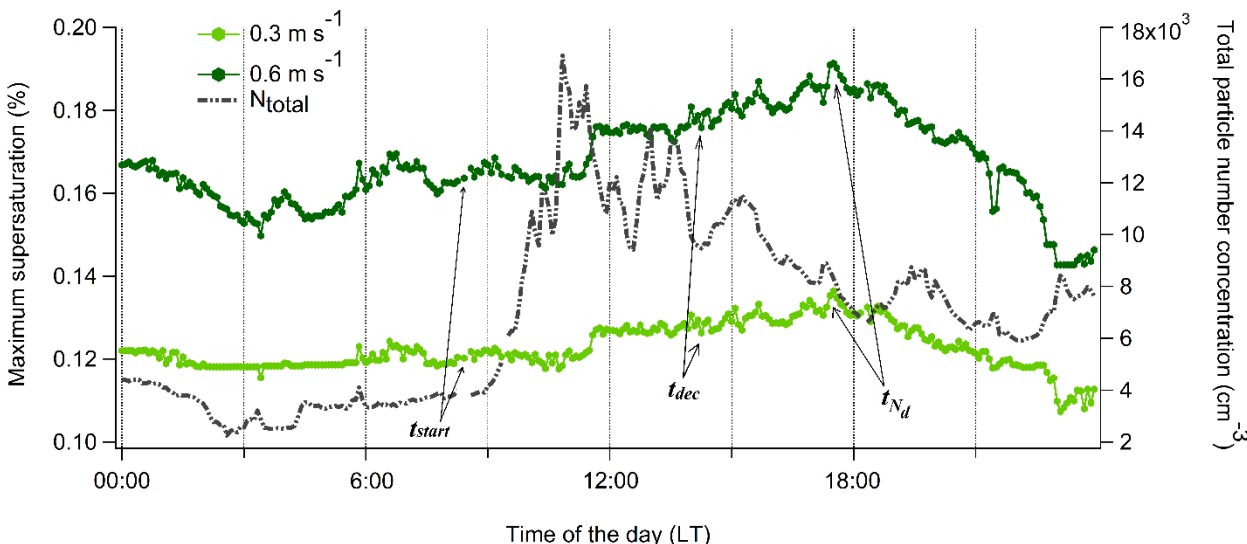

**Figure 6.** Diurnal evolution of the calculated maximum supersaturation ($s_{max}$) (lest axis) and total aerosol particle number concentrations ($N_{total}$) (right axis) for updraft velocities of $\sigma_w$= 0.3 m s$^{-1}$, and $\sigma_w$=0.6 m s$^{-1}$ during the "representative" new particle formation event captured at Finokalia on 29 August 2012.