# Peer review of "Regional New Particle Formation as Modulators of Cloud Condensation Nuclei and Cloud Droplet Number in the Eastern Mediterranean"

_Atmospheric Chemistry and Physics, 2018_

## Referee Comment (RC1) · Anonymous Referee #1 · 14 Dec 2018

This is a very comprehensive and carefully-made analysis on the influence of atmospheric new particle formation (NPF) on CCN concentrations and eventually on cloud droplet number concentrations (N) in the Mediterranean atmosphere. The authors introduce a new approach to estimate CCN production from NPF, and then use model simulations to get N. The paper is scientifically sound and well written. There a few incorrect statements in the paper, and a few places that require further discussion. I consider, however, these issues minor, since they do not require major effort or changes in writing of the text.

Important scientific issues

[Figure]

One main findings stressed by the authors is the suppressed effect of NPF on cloud droplet number concentration because the maximum supersaturation reached in a cloud updraft is lower at higher CCN concentrations. There are at least two things related to this point that should be discussed, or at least mentioned briefly, in the paper:

First, the non-linear response of the cloud droplet number concentration (N) to the CCN concentration, or to any bulk property representing the amount of aerosol particles, is a well-known feature reported in a number of model studies investigating cloud droplet activation, as well as in several field measurements.

Second, practically all cloud properties (albedo, probability of rain formation etc.) are expected to become more or less saturate at high concentrations of CCN (to some extend also at high N). This means an increase of the CCN concentration by a certain factor matters more in cleaner air. Since in most environments NPF is favored by low pre-existing particle concentration (i.e. cleaner air), this further means that the influence of NPF on cloud properties is usually expected to be greater than the influence of primary particle pollution in dirtier air.

Third, the authors correctly point out the assuming a constant cloud supersaturation biases the estimated influence of CCN (and hence NPF) on N. However, they come to this conclusion by assuming a constant cloud updraft velocity w (or its dispersion). The magnitude of w certainly depends on environmental conditions. This means, for example, that while it is not fair to assume a constant cloud supersaturation, it may also not to be fair to compare different seasons by assuming the same w at every season.

The authors estimate that NPF contributes to 39-69 % of the CCN budget in the supersaturation range 0.38-1 %. It should be noted their approach (as all the available approached based on field measurements) is only able to count on the influence of NPF on CCN if the newly-formed particle reach CCN size within less than a day or so after NPF. It is very likely that there are newly-formed particles that grow slower and still

survive to become CCN later on. So, the real contribution of NPF to the CCN budget is likely to be somewhat higher than the numbers obtained from this analysis. This issue is worth to be mentioned in the paper.

Minor and technical issues

Please use the term "cloud droplet number concentration" instead of "cloud droplet number" throughout the paper.

lines 67-70: Compared with Kulmala and Kerminen (2008), the topics of these lines are covered in much more detail the recent review by Kerminen et al (2018, Environ. Res. Lett. 13, 103003, https://doi.org/10.1088/1748-9326/aadf3c). The older review could be replaced with the newer one here.

line 73: Sipila et al. (2016, Nature, 537, 532-534, doi:10.1038/nature19314) provide the most comprehensive mechanistic description of coastal NPF presented so far. The authors might consider adding that reference here.

lines 91-92: please mention explicitly that dc refers to a critical diameter.

line 296: cloud have both updrafts and downdrafts, so I am sure that "cloud vertical velocity" is the proper wording here. The authors should maybe stick with "cloud updraft velocity" here as done elsewhere in the paper.

lines 372-378: It took me a while to understand the message of this discussion. The main point appears to be that hygroscopic properties (kappa) of a particle population tends to depend on the particle size, but this feature is not revealed by bulk aerosol composition measurements like those done with ACSM. And that this has consequences in interpreting the data. Please consider making the text a bit easier to follow.

lines 380-383: This statement is incorrect. There are at least 2 long-term studies in which the contribution of NPF to CCN has been investigated using direct CCN measurements (Sihto et al. 2011 already cited in this paper, Dameto de Espana et al.

[Figure]

2017, Atmos. Environ. 164, 289-298), not just particle number size distribution measurements.

line 396: "intermediate ions" is a commonly-used concept. What do the authors mean by "intermediate nucleation mode particles"?

lines 397-407: It is said that NPF starts at 8:30 and that these particles reach 100 nm at 21:30. This is not consistent with the given growth rate of 3.7 nm/h for nucleation mode particle. Does this mean the this particle population actually grows faster when reaching larger size, as mentioned in some other context later in the paper?

line 522: I suppose one of these velocities should be 0.6 m/s.

If Figures 3a and 3b are top of each other, it would be nice if their time axis matched with each other.

---

## Referee Comment (RC2) · Anonymous Referee #2 · 17 Jan 2019

The work by Kalkavouras et al. (2018) describes new metrics to evaluate the impact of new particle formation (NPF) on cloud condensation nuclei (CCN) budget and on cloud droplet number concentration (CDNC). The manuscript combines the analysis of an extended and valuable dataset, including both particle size distribution and chemical composition measured over 7 years at Finokalia, as well as model simulations to address the aspects related to CDNC. This manuscript aroused my interest and I believe it is worth publishing after some revisions. In its current form, there are several areas of the manuscript that need to be clarified, and in other areas the reader would benefit from additional information. Also, I believe that there are some inconsistencies between the different sections, and I think the authors contradict themselves in several areas of the manuscript. Finally, the distribution of information between the main text and the supplementary is sometimes questionable, and might be re-considered. In specific, CCN calculations performed at lower supersaturations (0.1%), which are expected to be more representative of real clouds, should be discussed in the main text. My detailed comments are listed below; they mainly concern the main text but the authors are encouraged to also take them into account to revise the abstract and the supplementary.

Comment 1: L55 & 58: It should be mentioned that Hyytiälä is located in Finland, in the boreal forest, and that the observatory of Chacaltaya is in Bolivia, at 5240 m a.s.l.

Comment 2: L70: Recent study by Kerminen et al. (2018) should be cited. Also, even if the present work does not aim at providing an exhaustive review of studies dedicated to marine environment, the papers by Sipilä et al. (2016) and Sellegri et al. (2016), which highlight the role of iodine in NPF, should be cited as well.

Comment 3: L92: The acronym $d_c$ should be explicitly defined.

Comment 4: L109-113: For consistency, it should be mentioned which locations are investigated in the paper by Kerminen et al. (2012). Also, the sentence from L110 to 113 should be checked carefully, as it is confusing (I would suggest to remove the last part "and the maximum … during an event").

Comment 5: L123: What do the "climate-relevant properties" refer to?

Comment 6: L129-133: The sentence should be rephrased.

Comment 7: L136-138: As suggested, this observation has already been reported, and should thus be supported by a reference. For instance, the paper by Leaitch et al. published in 1986 reported such observation.

Comment 8: L141-146: I would suggest to split the sentence into two parts, as it is too long in its current form. Also "depending" should be used instead of "depended", and "reported" instead of "reporting".

Comment 9: L146-149: It was thus already known/reported from the previous study by Kalkavouras et al. (2017) that discrepancies between CCN and CDNC enhancement arose partly from the supersaturations used for CCN calculations, which were too high compared to actual supersaturations observed in clouds. I thus wonder why, based on this result, the authors did not focus more on the CCN calculations performed at lower supersaturations (0.1%), which are discussed only in the supplementary.

Comment 10: L152-153: "continuous measurements of aerosol number size distributions and chemical composition": based on the information provided in Section 2.2, the chemical composition of the particles was not measured between January 2011 and April 2012, is that correct? If so, the expression "continuous" should be removed from the introduction, before more information is provided on data availability in the next sections.

Comment 11: L155: "characterize the differences between nucleated particles": what does that mean?

Comment 12: L158: "we consider all the issues": I think this is too strong. As an example, if the sensitivity of CDNC calculation to updraft velocities is partly investigated with the use of two different values, the seasonality of such parameter and related effect on predicted CDNC is not discussed. I would thus recommend to use a more balanced expression, or at least to remove "all".

Comment 13: L181-184: Please check the sentence; last part from "and thereafter…" is confusing.

Comment 14: L192-195: There is a word missing in the current form of the sentence: "sudden …?… of". Also, I would suggest to clearly mention the particle growth process: "by a sudden increase of nucleation-mode particles concentration (…), and further growth of these freshly formed particles that lead to a continuous increase in larger…"

Comment 15: L197-200: From what I understand, the method reported here is not consistent with the equations 1-3. Indeed, based on these equations, the width and location of the three modes (nucleation, Aitken and accumulation) are kept constant (9-25 nm, 25-100 nm and 100-848 nm), and the particle concentration in each mode is calculated from the sum of the particle concentration in all the size bins of the corresponding diameter range. How does this relate to the use of a multi log-normal distribution function? Was this method used in a first approach to get the "average" diameter ranges which are used in this work? This needs to be clarified.

Comment 16: L205: I would suggest to slightly change the wording to "$i_9$, $i_{100}$, $i_{848}$ refer to the SMPS size bins with mean (?) diameter 9, 100 and 848 nm, respectively".

Comment 17: L216-218: The knowledge of the PM1 chemical composition is a key parameter in the present work. I would thus recommend to briefly recall the method from Bougiatoti et al. 2009. In specific, the limits/uncertainties associated to this method, and how they affect the calculation of CCN and CDNC should be discussed. Also, when estimating the organic fraction, which ratio of OM/OC was used?

Comment 18: L218-223: More information about ACSM measurements and data analysis should be provided:

- What type of ACSM was used (Quad/Tof)?
- Standard/capture vapourizer?
- Did you apply any collection efficiency correction?

Comment 19: L232: I would write "a top-bottom column temperature difference", as if I am not mistaken (and even if it is quite straightforward!) the acronym T (and thus ΔT) has not been explained before in the text.

Comment 20: L236: What does "classified ammonium sulfate aerosol" mean?

Comment 21: L251: The equation should be given a number. Also I have several questions about the use of this equation:

- From what I understand, the main reason why to use this equation instead of a fixed $d_c$ is because it takes into account the chemical composition of the particles *via* kappa. However, when only filter measurements were available, there was only one kappa value available per day, right? Wasn't it so then that using the equation was in the end was very similar to using a constant $d_c$, as done in numerous previous studies?

- The variations of kappa appear to be quite limited on Fig. S3, so it is questionable how kappa actually affect $d_c$, and in the end, to which extent using the abovementioned equation improves the calculation of CCN concentration compared to the use of several fixed $d_c$. In other words, did you study, for a given supersaturation, the variations of $d_c$ caused by the variations of kappa?
- In connection with comment 17, did you evaluate the uncertainty on kappa calculation arising from the use of PM10 chemical composition to derive information about PM1? Did you evaluate the "magnitude" of the uncertainty on CCN calculation related to the use of these indirect measurements (couple with the fact that the size dependence of kappa is not taken into account) compared to that of the older method, with "reasonable" fixed $d_c$?

I would at least suggest to clearly mention the uncertainty/limits of the method which are highlighted in the previous questions, and/or better emphasize the benefits that I may have missed!

Comment 22: L290-291: In connection to my previous comment: would it be possible, for each supersaturation, to get an average $d_c$ from the CCN chamber measurements, then calculate the CCN concentrations corresponding to these "fixed" $d_c$ (in a similar way as done in the previous studies) and finally evaluate the corresponding prediction error? This would, in my view, help to assess the benefit from introducing the kappa in the CCN calculation, as suggested in the present work, or at least give an idea of the "limits" of this approach.

Comment 23: L299-300: In connection with comment 9: "determine the cloud-relevant supersaturations for which CCN perturbation calculations are relevant". Such "relevant supersaturations" have already been reported by Kalkavouras et al. (2017), so, again, calculations performed at 0.1% should in my view be the main focus of this work, and should be used to further link/compare CCN and CDNC results obtained in sections 3.2 and 3.3.

Also, did the author get the chance to evaluate the relevance of the predicted $N_d$ against for instance airborne in-situ measurements conducted in the vicinity of Finokalia?

Comment 24: L316: "vigorous boundary layer": do the author mean "turbulent"?

Comment 25: L332-336: The sentence should be checked and rephrased; also "were used" (L334) should be changed to "was used".

Comment 26: L344-354: Few suggestions:

- I would recall the periods during which each measurement technique was used;
- Wouldn't it be possible to summarize all the values on a plot, using pie charts for instance?
- It is not clear to me which instrument was used to derive the seasonal values discussed from L351 to 354;
- I was surprised to read that highest organic contribution was observed during wintertime; I would be curious to learn about the main sources during this time of the year.

Comment 27: L358-359: I do not understand this sentence: in my view the absolute concentrations should not affect kappa, only the fractions (i.e. "epsilon") should matter.

Comment 28: L364-368: The decrease of kappa between 6:00 and 9:00 LT is not obvious for me on Fig. S3… Also, would it be possible to add to Fig. S3 the time series of sulfate and organics measured with the ACSM, to support the hypotheses regarding the variations of kappa?

Comment 29: L370: Is the kappa difference of 0.2 kappa units calculated from average values? Because from Fig. S3, it seems that the difference can reach almost 0.4 (during the night and at the end of the afternoon).

Comment 30: L373-374: In connection with comment 28, the convergence which is reported in this sentence is also not obvious for me.

Comment 31: L374-378: This sentence is confusing me a lot, as, in my view, it conflicts with some ideas which are discussed elsewhere in the paper:

- "This constant character of the chemically derived kappa may be an evidence that using prescribed levels of supersaturation or critical diameters to calculate CCN concentrations can provide a biased influence of NPF events on CCN". In my view, the observation of the constant character of kappa does not indicate at this stage that the CCN predictions obtained from prescribed levels of supersaturation or prescribed diameters are biased; it only highlights the fact that both approaches are finally very similar, since the limited variations of kappa lead in the end to almost prescribed $d_c$. This assertion is even more surprising that Fig. S2 and L281-294 highlight a pretty good agreement between CCN prediction from ACSM/SMPS data and direct CCN measurements.
- "since there is a clear dependence between the chemical composition and the size of a particle": isn't is conflicting with L275-276 ("a size dependant consideration of hygroscopicity is therefore deemed unnecessary") and end of Section 2.4?

Comment 32: L386-389: The phrasing of the sentence is quite confusing; also, I wonder if it is relevant to apply this classification, which is exclusively based on measurements conducted in Pittsburgh, to measurements conducted at Finokalia, where particle concentrations and NPF event characteristics are most likely different. I would remove the sentence because I don't think it provides valuable information.

Comment 33: L396: What does "intermediate nucleation mode particles" mean?

Comment 34: L411: "the time series of the aerosol size distribution and chemical composition": since different datasets/instruments/measurement techniques were involved in this work, I would clearly recall that when filter measurements were used, there was only one kappa value available per day, to keep the message as clear as possible.

Comment 35: L415: I think there is a space missing: "supersaturations" -> "supersaturation $s$"

Comment 36: L417: Information in the brackets is not useful.

Comment 37: L420: It might be useful to also indicate $t_{start}$ on Fig. 3a. Also, the expression "Prior to 8:30 LT and 5 hours later" is not clear to me.

Comment 38: L429: "dividing" instead of "diving"?

Comment 39: L431: Over which period was the average value calculated? Full day?

Comment 40: L436-437: "from the influence of NPF on the larger supersaturations": even if I get the message, I think it would be more correct to change the wording to something like "from the influence of NPF on the production of particles which activate at larger supersaturations".

Comment 41: L438: On the 29[th] of August, the influence of NPF on CCN production is said to terminate at 21:30; was it decided that this "end time" would systematically be identified on the day of the NPF

event, or did the authors extended their research period to the next day, to document growth processes possibly spanning on several days?

Comment 42: L445-446: "this variation of $R_s$ can be equivalent to the percentage contribution of CCN owing to NPF": I would suggest to change to wording to something like "This variation of Rs indicates, for each supersaturation value, the increase of the CCN concentration related/due to particles originating from NPF".

Comment 43: L453: "the time which the new particles after the $t_{start}$ are able to grow": I would suggest to rephrase this part of the sentence to make it clearer.

Comment 44: L454-455: "This time fluctuates from 2.7 to 10.5 h in the 1.0-0.38%". The value of 2.7 h was obtained assuming an initial diameter of 25 nm for the newly formed particles at $t_{start}$, is that right? If so, I am not convinced by this approach, since I would expect most of the particles in the nucleation mode to be smaller than this upper limit at $t_{start}$.

Also, how did the authors get the upper value of 10.5 h? For me the longest time delay should correspond to particles with a diameter of 9 nm at $t_{start}$, which then need to reach 67 nm to be able to act as CCN at s = 0.38%, i.e. + 58 nm. Considering a GR of 3.7 nm/h, I find that it takes approximatively 7 hours for the particles to reach this $d_c$. Repeating the same calculation with initial diameter of 25 nm leads to a bit more than 11 hours. The hypotheses used for this calculation should be clarified.

Comment 45: L456: "start to feel the influence of NPF". In connection to comment 40, I also get the message but I would rephrase this part of the sentence, and refer more to the time it takes for the newly formed particles to grow to $d_c$ = 67 nm.

Comment 46: L456-458: "$t_{dec}$ is later for supersaturations below 0.7%": I do not understand the meaning of this sentence, since based on the definition from L425-429, there is one single value of $t_{dec}$ per event, which is derived from all supersaturations. Do the authors mean that it takes longer time to observe the influence of NPF on the concentration of particles able to act as CCN at lower supersaturations, as it is expected that those need to grow to larger sizes? Also, the link with the second part of the sentence is not clear to me.

Comment 47: L461-462: "indicating that the newly formed particles in this size range may exhibit similar chemical composition". Similar chemical composition at all sizes was assumed from the beginning of the calculation with the use of a single kappa for all sizes, wasn't it?! (L275).

Comment 48: L464-466: How did the authors get the reported values? By averaging all $R_s$ between 13:30 and 21:30? If so, in connection to comment 42, I would again talk more about an increase of the CCN concentration due to NPF rather than a contribution of NPF, and I would suggest to further check this aspect throughout the manuscript.

Comment 49: L466-467: "since the $d_c$ … respectively": I do not think this is a proper explanation, I would rather say "consistent with similar $R_c$ observed in the same size range, as mentioned above".

Comment 50: L469-476: I am not convinced by the suggested correction process for the two main reasons which are developed below:

- Background particles possibly contributing to CCN population together with growing particles originating from NPF are those which were already large enough before $t_{start}/t_{dec}$, not those in the nucleation mode before $t_{start}$. And, by the way, particle concentration in the nucleation mode should be around zero before $t_{start}$, as by definition those particles originate from nucleation.

- Also, if I am not mistaken, this paper discusses the CCN concentration increase from a reference concentration taken at $t_{start}$, which, I expect, already includes some contribution of background particles. I would thus say there is no need to apply any correction. The only bias, which is complex to evaluate but should still be mentioned, is caused by the possible appearance of large particles not originating from NPF between $t_{start}$ and end of NPF influence on CCN concentration (21:30 in this case), as those can impact the variations of the CCN concentration predicted at a certain (most likely low) supersaturation during this time period.

Comment 51: L478: I would suggest to remove "to the $R_s$ and subsequently".

Comment 52: L488: "for" should be removed.

Comment 53: L486-491: I would have expected particle GR to be the main factor determining the time delay between $t_{start}$ and $t_{dec}$, but the seasonal variation of this time delay (similar in winter, spring and summer, and lower compared to autumn) is not consistent with that of the GR reported by Kalivitis et al. (2018) (higher GR in summer, lower in winter and spring). Could the author comment on this aspect?

Comment 54: L494-495: "at cloud supersaturations encountered in this environment". To me, this sentence conflicts with what I think is a main message of the authors, yet already reported by Kalkavouras et al. (2017): L563, "the actual cloud supersaturation being much lower than the prescribed levels in the CCN analysis". (Also L33)

Comment 55: L496: Should be 3.3 instead of 3.4.

Comment 56: L504: I think the time interval between 8:30-11:00 is not correct to refer to the "growth hours" of the episode. This is at least not consistent with the fact that the influence of growing newly formed particles on CCN population is seen between 13:30 and 21:30.

Comment 57: L508: Space missing between "increases" and "4.7".

Comment 58: L509-515: "Both trends are related to decreases in accumulation mode aerosol number": the decrease of accumulation mode particle concentration is not clearly visible on Fig. 1. Also, I wonder why at this stage of the analysis the variations of $N_d$ are related to accumulation mode particles only, since the results of the previous section suggest a major contribution of Aitken mode particles to CCN population ($d_c < 67$ nm). To support their assumption, the authors would first need to discuss the inconsistency between the supersaturations used to predict CCN concentrations and $s_{max}$ retrieved by the model; $s_{max}$ being lower, it implies that particles need to grow to larger sizes to effectively act as CCN, and yes, in the end they most likely belong to accumulation instead of Aitken mode.

« as the latter has not had the chance to influence particles that act as CCN in clouds": again, I think this is not correct at this stage of the study, since the previous section highlights the influence of NPF on CCN concentration already from $t_{dec}$, i.e. 13:30. The supersaturation inconsistencies recalled above are also needed to further explain/clarify this aspect. Particles can act at CCN already from $t_{dec}$, but in the presence of supersaturations which are most likely significantly higher than those predicted by the model.

Considering the different supersaturations discussed in sections 3.2 and 3.3, I think it is finally complex to establish/comment on the link between $t_{dec}$ and $t_{nd}$.

Another point: how do the authors determine the beginning of NPF influence on $N_d$ at 17:25? From Fig.5, it seems that the most significant increase of $N_d$ is seen from ~21:00. Now looking at Fig. 1.A, this time coincides with the time at which the NPF event is somewhat interrupted, suggesting that the

particles contributing to the increase of $N_d$ could finally not be related to NPF. Could the authors comment on that?

Comment 59: L519-529: It is complex for me to understand this part of the analysis. Specific comment on L527, "Since $N_d$ does not increase significantly until midnight": looking at Fig. 5 I would say it does, at least until 23:00. Second part of the sentence is also not clear to me.

Comment 60: L533: I do not understand how the authors calculate the value of 30 cm$^{-3}$.

Comment 61: L534: In connection to comment 41, why is the analysis stopped at midnight? Again, if the analysis is limited to the day of the event, this has to be mentioned, also the reason why.

Comment 62: L535-538: I think the length of the dataset is a strong point of this work, so I would really suggest to discuss more the "statistics" in the main text. In general, the supplementary includes valuable information, from which the reader would benefit more if it was partly moved to the main text. This comment also applies to CCN related calculations reported in the previous section (see for instance comments 9 and 23).

Comment 63: L548: "accurately" is in my view too strong.

Comment 64: L550, 553: time delays between $t_{start}$ and $t_{dec}$ are different from those reported in Sect. 3.2. Also, it would be better to have a uniform notation to report durations (L550, 553 and 554).

Comment 65: L558-559: "the impact of NPF on $N_d$ differs considerably from the CCN based analysis". I do not completely agree with this sentence, as it compares two different variables (CCN concentration and $N_d$) calculated using different hypotheses.

Comment 66: L559: "Regardless of season". This assessment is a bit too strong in my view, as a possible seasonal variation of some parameters such as vertical velocity (and thus further effect on $N_d$ calculation) has not been discussed. Also, I would not talk about "typical boundary layer clouds"; L310 indicates "in cloudy boundary layer in the region".

Comment 67: L569-570: Please refer to comment 50.

Comment 68: L571-575: I would not say this is a striking consequence; I would rather say it is expected, particles need to grow to large-enough sizes to be activated into droplets, which takes time, and this is even more the case when supersaturations get lower. I am also not sure about the last part of the first sentence "$N_d$ is insensitive to increase in CCN during the course of an event… vapor": again, during the course of the event, CCN population possibly activating at higher supersaturation is increased first, and as particles are getting bigger towards the end of the event, they can activate at lower supersaturations.

Comment 69: L579-580: Wasn't it already a result from Kalkavouras et al. (2017)? See also comments 9 and 23.

Comment 70: L581: The influence of what?

Comment 71: L584: "highly effective paradigm" is too strong for me; if new metrics are introduced for quantifying the temporal aspect of NPF influence on CCN production, CCN calculation itself relies on the use of prescribed supersaturation, which is simultaneously reported to be hazardous (eg L525, 581).

Comment 72: L741-749: Check alphabetical order.

Comment 73: Fig. 1: Caption, (b), I would remove "differences", and simply refer to the time series of the particle concentration at different time resolutions.

Comment 74: Fig. 3: it would help to have similar scale on the x axis of Fig. 3 a and b.

Comment 75: Fig. S1: x axis label: "ACSM" instead of "ASCM". Also, adding the 1:1 line would help to interpret the results.

Comment 76: Fig. S3: Caption: space missing between "the" and "kappa".

Comment 77: Table S2: I would suggest to clearly explain what "bef" and "aft" refer to, as these are used in several tables of the supplementary. Also, in the caption, it is not clear to which "relative contribution" the authors are referring to. I would clearly mention it is the increase of Rs (%) observed after $t_{dec}$, and clearly indicate it corresponds to the "Change" column.

Comment 78: Table S5: If I am not mistaken, the authors never refer to Table S5, neither in the main text nor in the supplementary. This table report values which would have been very interesting to discuss more in the main text, in specific those calculated for s = 0.1%, as this supersaturation is thought to be more "representative" of real clouds. Also, more information would be needed regarding the calculation of the $N_{CCN}$ values, as I think it has not been explained elsewhere: over which time period (average between $t_{dec}$ and end of NPF influence?)? Any correction applied?

Comment 79: Supplementary: several spaces missing in different places.

*Bibliography*

Bougiatioti, A., Fountoukis, C., Kalivitis, N., Pandis, S. N., Nenes, A., and Mihalopoulos, N.: Cloud condensation nuclei measurements in the marine boundary layer of the Eastern Mediterranean: CCN closure and droplet growth kinetics, Atmos. Chem. Phys., 9, 7053-7066, https://doi.org/10.5194/acp-9-7053-2009, 2009.

Kalivitis, N., Kerminen, V.-M., Kouvarakis, G., Stavroulas, I., Tzitzikalaki, E., Kalkavouras, P., Daskalakis, N., Myriokefalitakis, S., Bougiatioti, A., Manninen, H. E., Roldin, P., Petäjä, T., Boy, M., Kulmala, M., Kanakidou, M., and Mihalopoulos, N.: Formation and growth of atmospheric nanoparticles in the eastern Mediterranean: Results from long-term measurements and process simulations, Atmos. Chem. Phys. Discuss., https://doi.org/10.5194/acp-2018-229, in review, 2018.

Kalkavouras, P., Bossioli, E., Bezantakos, S., Bougiatioti, A., Kalivitis, N., Stavroulas, I., Kouvarakis, G., Protonotariou, A. P., Dandou, A., Biskos, G., Mihalopoulos, N., Nenes, A., and Tombrou, M.: New particle formation in the southern Aegean Sea during the Etesians: importance for CCN production and cloud droplet number, Atmos. Chem. Phys., 17, 175-192, https://doi.org/10.5194/acp-17-175-2017, 2017.

Kerminen, V.-M., Paramonov, M., Anttila, T., Riipinen, I., Fountoukis, C., Korhonen, H., Asmi, E., Laakso, L., Lihavainen, H., Swietlicki, E., Svenningsson, B., Asmi, A., Pandis, S. N., Kulmala, M., and Petäjä, T.: Cloud condensation nuclei production associated with atmospheric nucleation: a synthesis based on existing literature and new results, Atmos. Chem. Phys., 12, 12037-12059, https://doi.org/10.5194/acp-12-12037-2012, 2012.

Kerminen, V.-M., Chen, X., Vakkari, V., Petäjä, T., Kulmala, M., and Bianchi, F.: Atmospheric new particle formation and growth: review of field observations, Environ. Res. Lett., 10, https://doi.org/10.1088/1748-9326/aadf3c, 2018.

K. Sellegri, J. Pey, C. Rose, A. Culot, H.L. DeWitt, S. Mas, A.N. Schwier, B. Temime-Roussel, B. Charriere, A. Saiz-Lopez, A. S. Mahajan, D. Parin, A. Kukui, R. Sempere, B. D'anna, and N. Marchand, Evidence of atmospheric nanoparticle formation from emissions of marine microorganisms, Geophys. Res. Lett., 43(12), 2016GL069389, doi:10.1002/2016GL069389, 2016.

W. R. Leaitch, J. W. Strapp and *G.* A. Isaac: Cloud droplet nucleation and cloud scavenging of aerosol sulphate in polluted atmospheres, Tellus, *388,* 328-344, 1986.

Mikko Sipilä, Nina Sarnela, Tuija Jokinen, Henning Henschel, Heikki Junninen, Jenni Kontkanen, Stefanie Richters, Juha Kangasluoma, Alessandro Franchin, Otso Peräkylä, Matti P. Rissanen, Mikael Ehn, Hanna Vehkamäki, Theo Kurten, Torsten Berndt, Tuukka Petäjä, Douglas Worsnop, Darius Ceburnis, Veli-Matti Kerminen, Markku Kulmala, and Colin O'Dowd : Molecular scale evidence of new particle formation *via* sequential addition of HIO3, Nature,  537(7621): 532–534, doi:10.1038/nature19314, 2016.

---

## Author Comment (AC1) · 4 Mar 2019

**Response to Reviewer #1:**

*This is a very comprehensive and carefully-made analysis on the influence of atmospheric new particle formation (NPF) on CCN concentrations and eventually on cloud droplet number concentrations (N) in the Mediterranean atmosphere. The authors introduce a new approach to estimate CCN production from NPF, and then use model simulations to get N. The paper is scientifically sound and well written. There a few incorrect statements in the paper, and a few places that require further discussion. I consider, however, these issues minor, since they do not require major effort or changes in writing of the text.*

The authors would like to thank the reviewer for the positive feedback and thoughtful and thorough comments. A point-by-point reply to all of the issues raised and the corresponding changes follows.

**Important scientific issues:**

*One main findings stressed by the authors is the suppressed effect of NPF on cloud droplet number concentration because the maximum supersaturation reached in a cloud updraft is lower at higher CCN concentrations. There are at least two things related to this point that should be discussed, or at least mentioned briefly, in the paper:*

*First, the non-linear response of the cloud droplet number concentration (N) to the CCN concentration, or to any bulk property representing the amount of aerosol particles, is a well-known feature reported in a number of model studies investigating cloud droplet activation, as well as in several field measurements.*

We completely agree! These points will be further clarified in the text, adding representative references over decades (e.g. Twomey et al., 1977; Ghan et al., 1993; Boucher and Lohmann, 1995; Gultepe, and Isaac, 1996; Nenes et al., 2001; Ramanathan et al., 2001; Ghan et al., 2011; Sullivan et al., 2016) that express this sub-linearity.

*Second, practically all cloud properties (albedo, probability of rain formation etc.) are expected to become more or less saturate at high concentrations of CCN (to some extend also at high N). This means an increase of the CCN concentration by a*

*certain factor matters more in cleaner air. Since in most environments NPF is favored by low pre-existing particle concentration (i.e. cleaner air), this further means that the influence of NPF on cloud properties is usually expected to be greater than the influence of primary particle pollution in dirtier air.*

The point is well taken. In the manuscript we make very clear that everything discussed refers to the conditions seen at the eastern Mediterranean, and that the background values prior to the NPF event determine the extent to which the additional CCN can affect cloud properties. These points will be further emphasized in the conclusions and abstract of the revised paper.

In response to the comments raised, we looked into the Finokalia datasets among the 162 days with NPF episodes and note the relationship between the pre-existing particle concentration and the CCN enhancement. Indeed, the increase of the CCN concentrations when NPF recorded under cleaner air conditions is slightly higher (on average ~7.5±5.3%) than in polluted days (on average 7.0±4.9%) and will be noted in the discussion and conclusions as well.

*Third, the authors correctly point out the assuming a constant cloud supersaturation biases the estimated influence of CCN (and hence NPF) on N. However, they come to this conclusion by assuming a constant cloud updraft velocity w (or its dispersion). The magnitude of w certainly depends on environmental conditions. This means, for example, that while it is not fair to assume a constant cloud supersaturation, it may also not to be fair to compare different seasons by assuming the same w at every season.*

We understand the reviewer's point. In the approach adopted here, the method itself could use actual observations of vertical velocity (w). This is not done here, because measurements of in-cloud vertical velocity were not available to use. That said, *a*) we use the method of "characteristic velocity" (Morales and Nenes, 2010), so that the droplet numbers calculate correspond to the average value over the probability distribution of positive vertical velocity; *b*) the marine boundary layer does not exhibit a considerably wider spectral dispersion higher than 0.3 m s$^{-1}$ ("base-case" characteristic velocity, reflecting summertime boundary layer conditions in the vicinity of Finokalia), especially when considering the sensitivity test, where the vertical velocity dispersion was doubled ($\sigma_w$=0.6 m s$^{-1}$). The above-mentioned

distribution of vertical velocity in the marine boundary layer, is consistent with vertical velocities observed in marine boundary layers (e.g. Albrecht et al., 1998 and references therein; Fountoukis et al., 2007; Ghan et al., 2011), where w displays a spectral dispersion around zero value ($\sigma_w$ is calculated to be between 0.2 and 0.3 m s$^{-1}$); $c$) the relative contribution of NPF to CDNC was similar between these cases (21.9±6.5%,). Given the above, and considering that seasonal changes in $\sigma_w$ do not exceed the range examined, we feel that the results presented here provide a fair representation of the seasonality of NPF effects on CDNC.

*The authors estimate that NPF contributes to 39-69% of the CCN budget in the supersaturation range 0.38-1%. It should be noted their approach (as all the available approaches based on field measurements) is only able to count on the influence of NPF on CCN if the newly-formed particle reach CCN size within less than a day or so after NPF. It is very likely that there are newly-formed particles that grow slower and still survive to become CCN later on. So, the real contribution of NPF to the CCN budget is likely to be somewhat higher than the numbers obtained from this analysis. This issue is worth to be mentioned in the paper.*

We thank the reviewer for pointing out this important issue. We have mentioned this point early in the introduction (see lines 79 – 83), and carefully stated that all the calculations are for "perturbations from fresh NPF events upon the background aerosol that in itself may have been shaped by NPF earlier on, (lines 470 – 482 in the main text)". We also note that the growth of preexisting particles (that some may be from earlier NPF events) also contribute to the droplet number and present an analysis that deconvolves their effect on $s_{max}$ and droplet number (see lines 520 – 531).

**Minor and technical issues:**

*Please use .. paper.*

Done.

*lines 67-70: Compared … replaced with the newer one here.*

Done.

*line 73: Sipila et al. (2016, Nature, … consider adding that reference here.*

We thank the reviewer for pointing this out. Done.

*lines 91-92: Please mention explicitly that $d_c$ refers to a critical diameter.*

Done.

*line 296: Cloud have both updrafts … in the paper.*

Done.

*lines 372-378: It took me … making the text a bit easier to follow.*

This section has now been revised for clarity.

*lines 380-383: This statement is incorrect. There ...size distribution measurements.*

Thank you for pointing this out. We will replace the word "all" with "several" and the relevant references will be accordingly added.

*line 396: "intermediate ions" is a commonly-used concept. What do the authors mean by "intermediate nucleation mode particles"?*

The "intermediate nucleation mode particles" term used throughout the manuscript, corresponds to particles with diameters from 9 to 25 nm. Since we had no means to determine the intermediate negative-ion concentrations, this concept can be used as a quality test, in order to determine the initiation of a NPF event. The starting of the NPF is further confirmed by the evolution of the particle size distribution ("banana-shape pattern) when the new 9–nm particles appear and shift gradually towards larger sizes. We will make this point very clear.

*lines 397-407: It is said that NPF starts at 8:30 and that these particles reach 100 nm at 21:30. This is not consistent with the given growth rate of 3.7 nm/h for nucleation mode particle. Does this mean that this particle population actually grows faster when reaching larger size, as mentioned in some other context later in the paper?*

This is a good point. We will include the respective reference in order to avoid this confusing point. Thus, we are going to revise the text as: "Afterwards, particles continued to grow faster in size (Paasonen et al., 2018) for several hours, exceeding 100 nm in diameter at 21:30 LT.".

*line 522: I suppose one of these velocities should be 0.6 m/s.*

Indeed so! Thank you for pointing this out.

*If Figures 3a and 3b are top of each other, it would be nice if their time axis matched with each other.*

We will modify it in the revised version according to your suggestion.

**References**

Albrecht, B.A.: Observations of cloudy boundary layers, Chapter 8, Verhandelingen Natuurkunde, Eerste Reeks, deel 48, 179-198, 1998.

Boucher, O., and Lohmann, U.: The sulfate-CCN-cloud albedo effect, Tellus 47B, 281-300, 1995.

Fountoukis, C., Nenes, A., Meskhidze, N., Bahreini, R., Conant, W. C., Jonsson, H., Murphy, S., Sorooshian, A., Varutbangkul, V., Brechtel, F., Flagan, R. C., and Seinfeld, J. H.: Aerosol-cloud drop concentration closure for clouds sampled during the International Consortium for Atmospheric Research on Transport and Transformation 2004 campaign, J. Geophys. Res., 112, D10S30, doi:10.1029/2006JD007272, 2007.

Ghan, S., Chung, C., and Penner, J.: A parameterization of cloud droplet nucleation part I: single aerosol type, Atmosph. Res., Vol. 30, Issue 4, 198-221, doi:10.1016/0169-8095(93)90024-I,1993.

Ghan, S., Abdul-Razzak, H., Nenes, A., Ming, Y., Liu., X., Ovchinnikov, M., Shipway, B., Meskhidze, N., Xu, J., and Shi, X.: Droplet nucleation: physically-based parameterizations and comparative evaluation, J. Adv. Model Earth Syst., Vol. 3, M10001, doi:10.1029/2011MS000074, 2011.

Gultepe, I., and Isaac, G. A.: The relationship between cloud droplet and aerosol number concentrations for climate models, International Journal of Climatology, Vol. 16, 941-946, 1996.

Morales, R. and Nenes, A.: Characteristic updrafts for computing distribution-averaged cloud droplet number and stratocumulus cloud properties, J. Geophys. Res., 115, D18220, doi:10.1029/2009JD013233, 2010.

Nenes, A., Chan, S., Abdul-Razzak, H., Chuang, P., and Seinfeld, J.: Kinetic limitations on cloud droplet formation and impact on cloud albedo, Tellus 53B, 133-149, doi:10.3402/tellusb.v53i2.16569, 2001.

Ramanathan, V., Crutzen, P. J., Kiehl, J. T., and Rosenfeld, D.: Aerosols, climate, and the hydrological cycle, Science 294 (5549), 2119-2124. doi:10.1126/science.1064034, 2001.

Sullivan, S., Lee, D., Oreopoulos, L., and Nenes, A.: Role of updraft velocity in temporal variability of global cloud hydrometeor number, PNAS, Vol. 113 (21), 5791-5796, doi:10.1073/pnas.1514039113, 2016.

Twomey, S.: The influence of pollution on the shortwave albedo of clouds, J. Atmos. Sci., 34, 1977.

---

## Author Comment (AC2) · 4 Mar 2019

**Response to Reviewer #2:**

*The work by Kalkavouras et al. (2018) describes new metrics to evaluate the impact of new particle formation (NPF) on cloud condensation nuclei (CCN) budget and on cloud droplet number concentration (CDNC). The manuscript combines the analysis of an extended and valuable dataset, including both particle size distribution and chemical composition measured over 7 years at Finokalia, as well as model simulations to address the aspects related to CDNC. This manuscript aroused my interest and I believe it is worth publishing after some revisions. In its current form, there are several areas of the manuscript that need to be clarified, and in other areas the reader would benefit from additional information. Also, I believe that there are some inconsistencies between the different sections, and I think the authors contradict themselves in several areas of the manuscript. Finally, the distribution of information between the main text and the supplementary is sometimes questionable, and might be re-considered. In specific, CCN calculations performed at lower supersaturations (0.1%), which are expected to be more representative of real clouds, should be discussed in the main text. My detailed comments are listed below; they mainly concern the main text but the authors are encouraged to also take them into account to revise the abstract and the supplementary.*

We would like to thank the reviewer for his/her positive and helpful comments which ameliorated the analysis of the manuscript. The reviewer's suggestion to include the CCN calculations at 0.10% supersaturation is now integrated into the main text, together with the relevant discussion. Please find below a point-to-point reply to each issue raised.

*Comment 1: L55 & 58: It should be mentioned that Hyytiala is located in Finland, in the boreal forest, and that the observatory of Chacaltaya is in Bolivia, at 5240 m a.s.l.*

Done.

*Comment 2: L70: Recent study by Kerminen et al. (2018) should be cited. Also, even if the present work does not aim at providing an exhaustive review of studies dedicated to marine environment, the papers by Sipila et al. (2016) and Sellegri et al. (2016), which highlight the role of iodine in NPF, should be cited as well.*

Done.

*Comment 3: L92: The acronym $d_c$ should be explicitly defined.*

Done.

*Comment 4: L109-113: For consistency, it should be mentioned which locations are investigated in the paper by Kerminen et al. (2012). Also, the sentence from L110 to 113 should be checked carefully, as it is confusing (I would suggest to remove the last part "and the maximum … during an event").*

Done.

*Comment 5: L123: What do the "climate-relevant properties" refer to?*

We thank the reviewer for raising this point. This was a typo, and the text should read: "aerosol-cloud interactions and climate-related responses".

*Comment 6: L129-133: The sentence should be rephrased.*

Done.

*Comment 7: L136-138: As suggested, this observation has already been reported, and should thus be supported by a reference. For instance, the paper by Leaitch et al. published in 1986 reported such observation.*

We thank the reviewer for the suggestion, and we will add representative references that express this sub-linearity (e.g. Twomey et al., 1977; Leaitch et al., 1986; Ghan et al., 1993; Boucher and Lohmann, 1995; Gultepe, and Isaac, 1996; Nenes et al., 2001; Ramanathan et al., 2001; Ghan et al., 2011; Sullivan et al., 2016).

*Comment 8: L141-146: I would suggest to split the sentence into two parts, as it is too long in its current form. Also "depending" should be used instead of "depended", and "reported" instead of "reporting".*

Done.

*Comment 9: L146-149: It was thus already known/reported from the previous study by Kalkavouras et al. (2017) that discrepancies between CCN and CDNC enhancement arose partly from the supersaturations used for CCN calculations, which were too high compared to actual supersaturations observed in clouds. I thus wonder why, based on this result, the authors did not focus more on the CCN calculations performed at lower supersaturations (0.1%), which are discussed only in the supplementary.*

We would like to thank the reviewer for this suggestion and agree that 0.10% supersaturation better reflects clouds in the area, something which is now included and discussed throughout the revised version of the manuscript. Supersaturations between 0.38 and 0.73% were included in the first place, since the measurements of CCN concentration ($cm^{-3}$) were performed at these levels and allowed closure calculations with the measured size distributions.

*Comment 10: L152-153: "continuous measurements of aerosol number size distributions and chemical composition": based on the information provided in Section 2.2, the chemical composition of the particles was not measured between January 2011 and April 2012, is that correct? If so, the expression "continuous" should be removed from the introduction, before more information is provided on data availability in the next sections.*

Indeed, particle chemical composition was not measured between January 2012 and April 2012. Thus "continuous" will be removed from the text.

*Comment 11: L155: "characterize the differences between nucleated particles": what does that mean?*

We refer to the timing properties from the beginning of NPF events (i.e. starting time ($t_{start}$) and the duration time interval) throughout their activation into cloud droplets ($t_{Nd}$ and duration time interval). We will make this point very clear.

*Comment 12: L158: "we consider all the issues": I think this is too strong. As an example, if the sensitivity of CDNC calculation to updraft velocities is partly investigated with the use of two different values, the seasonality of such parameter and related effect on predicted CDNC is not discussed. I would thus recommend to use a more balanced expression, or at least to remove "all".*

We have amended the point to "consider major issues". Regarding the comment on the importance of updraft seasonality (which we consider it to be of secondary importance), please see our relevant response to Reviewer #1.

*Comment 13: L181-184: Please check the sentence; last part from "and thereafter…" is confusing.*

Indeed, the last part of the sentence will be rephrased.

*Comment 14: L192-195: There is a word missing in the current form of the sentence: "sudden …?...of". Also, I would suggest to clearly mention the particle growth process: "by a sudden increase of nucleation-mode particles concentration (…), and further growth of these freshly formed particles that lead to a continuous increase in larger…"*

We apologize for this. "increase" is the missing word, and we also modified the sentence according to your suggestion.

*Comment 15: L197-200: From what I understand, the method reported here is not consistent with the equations 1-3. Indeed, based on these equations, the width and location of the three modes (nucleation, Aitken and accumulation) are kept constant (9-25 nm, 25-100 nm and 100-848 nm), and the particle concentration in each mode is calculated from the sum of the particle concentration in all the size bins of the corresponding diameter range. How does this relate to the use of a multi log-normal distribution function? Was this method used in a first approach to get the "average" diameter ranges which are used in this work? This needs to be clarified.*

The phrase: "The concentration of particles … as follows:" will be rephrased as: "The modal concentration of particles is obtained from each SMPS size distribution as follows:", since according to the equations 1-3, the width of the three modes are kept constant (9-25 nm for the nucleation mode, 25-100 nm for the Aitken mode, and 100-848 nm for the accumulation mode), and the particle number concentration in each mode is determined from the sum of the particle concentration in all the size bins of the corresponding diameter range.

*Comment 16: L205: I would suggest to slightly change the wording to "$i_9$, $i_{100}$, $i_{848}$ refer to the SMPS size bins with mean (?) diameter 9, 100 and 848 nm, respectively".*

Amended.

*Comment 17: L216-218: The knowledge of the $PM_1$ chemical composition is a key parameter in the present work. I would thus recommend to briefly recall the method from Bougiatioti et al. 2009. In specific, the limits/uncertainties associated to this method, and how they affect the calculation of CCN and CDNC should be discussed. Also, when estimating the organic fraction, which ratio of OM/OC was used?*

In Bougiatioti et al. (2009) bulk chemical composition from daily filter analysis was used, as was done for this study from 2008-2012. When water-solubility of organics is

considered, CCN closure was achieved with an error of 0.6±6%. Given that this study focuses on data from the same site, we have every reason to expect that the same approach can be used in this study as well. Limitations/uncertainties associated with this method remain the lack of size-resolved chemical composition, which will be clarified in the revised text.

From the available ACSM measurements, Organic Matter (OM) is directly derived and used for the calculation of the organic fraction. For the period when daily filter samples were used for the chemical composition, a ratio of OM/OC of 2.1 was used, based on other studies from the site (Sciare et al., 2005; Hildebrandt et al., 2010).

*Comment 18: L218-223: More information about ACSM measurements and data analysis should be provided:*
*- What type of ACSM was used (Quad/Tof)?*
*- Standard/capture vapourizer?*
*- Did you apply any collection efficiency correction?*

We would like to thank the reviewer for pointing out this omission. The ACSM measurements were performed with a Quadrupole ACSM, equipped with a standard vaporizer (Ng et al., 2011). The response factor (RF) for nitrate along with the relative ionization efficiencies (RIEs) for ammonium and sulfate were determined by ammonium nitrate and ammonium sulfate calibrations and the RIE for sulfate was determined according to the fitting approach proposed by Budisulistiorini et al. (2014). Mass concentrations were corrected using a chemical composition dependent collection efficiency (Middlebrook et al., 2012). All this information will be included in the revised version.

*Comment 19: L232: I would write "a top-bottom column temperature difference", as if I am not mistaken (and even if it is quite straightforward!) the acronym T (and thus ΔT) has not been explained before in the text.*

Amended.

*Comment 20: L236: What does "classified ammonium sulfate aerosol" mean?*

This is standard terminology in the aerosol literature, and it means "monodisperse" aerosol generated from a differential mobility analyzer (DMA). We will clarify this in the revised text.

***Comment 21: L251: The equation should be given a number. Also I have several questions about the use of this equation:***

***- From what I understand, the main reason why to use this equation instead of a fixed $d_c$ is because it takes into account the chemical composition of the particles via kappa. However, when only filter measurements were available, there was only one kappa value available per day, right? Wasn't it so then that using the equation was in the end was very similar to using a constant $d_c$, as done in numerous previous studies?***

We understand the reviewer's concern. Past experience has shown (e.g. Bougiatioti et al., 2009, 2011) that the composition displays remarkably consistent behavior, so filter measurements may indeed sufficiently constrain composition. The successful CCN closure shows that indeed our approach is sufficient. These points will be clarified in the revised text.

***- The variations of kappa appear to be quite limited on Fig. S3, so it is questionable how kappa actually affect $d_c$, and in the end, to which extent using the above mentioned equation improves the calculation of CCN concentration compared to the use of several fixed $d_c$. In other words, did you study, for a given supersaturation, the variations of $d_c$ caused by the variations of kappa?***

We kindly disagree that the variations of kappa shown in Fig. S3 are quite limited. For example, purple triangles, corresponding to supersaturations from 0.60 to 0.70% are more representative of particles with critical diameters from 47 to 69 nm, and have a kappa value of around 0.15, while blue circles that correspond to supersaturations <0.20% are more representative of particles with an average critical diameter of 130 nm and have a kappa value of around 0.4. Moreover, for a given supersaturation (e.g. 0.20%) for the specific studied day kappa values vary from 0.22 to 0.51 corresponding to $d_c$ variations from 85 to 115 nm.

***- In connection with comment 17, did you evaluate the uncertainty on kappa calculation arising from the use of $PM_{10}$ chemical composition to derive information about $PM_1$? Did you evaluate the "magnitude" of the uncertainty on CCN calculation related to the use of these indirect measurements (couple with the fact that the size dependence of kappa is not taken into account) compared to that of the older method, with "reasonable" fixed $d_c$?***

*I would at least suggest to clearly mention the uncertainty/limits of the method which are highlighted in the previous questions, and/or better emphasize the benefits that I may have missed!*

We agree that the above concerns are valid in the general sense. It's been established, however, that the aerosol in Finokalia has most of its $PM_{10}$ hygroscopicity in the $PM_1$ fraction. Long-term studies (e.g. Koulouri et al., 2008; Bougiatioti et al., 2013) at the site have established that sulfate is by majority found in the fine fraction (82.7±12.7% of $PM_{10}$ sulfate found in $PM_1$) and the same applies also for ammonium (88±13.3% of $PM_{10}$ ammonium found in $PM_1$). 75±11% of $PM_{10}$ organic matter is also found in $PM_1$. A sensitivity calculation demonstrates that the resulting uncertainty on the kappa considering these major components is about 2.5±0.2%, which has an almost insignificant impact on the calculated CDNC. This discussion, and other clarifications, will certainly be added in the revised version.

*Comment 22: L290-291: In connection to my previous comment: would it be possible, for each supersaturation, to get an average $d_c$ from the CCN chamber measurements, then calculate the CCN concentrations corresponding to this "fixed" $d_c$ (in a similar way as done in the previous studies) and finally evaluate the corresponding prediction error? This would, in my view, help to assess the benefit from introducing the kappa in the CCN calculation, as suggested in the present work, or at least give an idea of the "limits" of this approach.*

From the available CCN data we calculated a mean $d_c$ at each supersaturation level, and afterwards estimated the CCN number concentrations for the respective "fixed" $d_c$. Using both the calculating CCN from a "fixed" $d_c$ against the CCN concentrations from chemical composition and size-distribution measurements, we assessed the two different approaches at 0.20, 0.38, 0.52, 0.73 and 1.00% supersaturation, respectively. The values of our initial approach with estimated CCN concentrations from kappa and size-distribution measurements are generally higher. More specifically, when using a "fixed" $d_c$ estimated CCN concentrations are almost 30% lower compared to the respective ones when using kappa and the size-distribution measurements for all supersaturation levels above 0.38%. For 0.20% supersaturation, estimated CCN concentrations are approximately 60% lower for the "fixed" $d_c$ approach. This point will be noted in the text, according to your suggestion.

*Comment 23: L299-300: In connection with comment 9: "determine the cloud-relevant supersaturations for which CCN perturbation calculations are relevant". Such "relevant supersaturations" have already been reported by Kalkavouras et al. (2017), so, again, calculations performed at 0.1% should in my view be the main focus of this work, and should be used to further link/compare CCN and CDNC results obtained in sections 3.2 and 3.3.*

The CCN calculations at supersaturation 0.10% will be added in the revised text.

*Also, did the author get the chance to evaluate the relevance of the predicted $N_d$ against for instance airborne in-situ measurements conducted in the vicinity of Finokalia?*

Unfortunately, there were no in-situ measurements of cloud droplet number concentration at Finokalia to evaluate against calculated $N_d$. Studies in the past, however, using the same parameterization have proven quite successful (e.g. Fountoukis et al., 2007; Morales et al., 2010) – and there is no reason to think that the same level of performance would be any different here.

*Comment 24: L316: "vigorous boundary layer": do the authors mean "turbulent"?*

Indeed, thus we will replace the word "vigorous" according to your suggestion.

*Comment 25: L332-336: The sentence should be checked and rephrased; also "were used" (L334) should be changed to "was used".*
Done.

*Comment 26: L344-354: Few suggestions:*
*- I would recall the periods during which each measurement technique was used;*

Done.

*- Wouldn't it be possible to summarize all the values on a plot, using pie charts for instance?*

We thank the reviewer for the suggestion. Pie charts for $PM_1$ will be included to summarize the chemical composition of submicron particulate matter throughout the NPF episodes.

*- It is not clear to me which instrument was used to derive the seasonal values discussed from L351 to 354;*

This is now clarified.

*- I was surprised to read that highest organic contribution was observed during wintertime; I would be curious to learn about the main sources during this time of the year.*

The differences are truly minimal; we hypothesize that during winter, long-range transport of organic-rich material from the Greek mainland may be responsible.

*Comment 27: L358-359: I do not understand this sentence: in my view the absolute concentrations should not affect kappa, only the fractions (i.e. "epsilon") should matter.*

Indeed so. This is now corrected.

*Comment 28: L364-368: The decrease of kappa between 6:00 and 9:00 LT is not obvious for me on Fig. S3... Also, would it be possible to add to Fig. S3 the time series of sulfate and organics measured with the ACSM, to support the hypotheses regarding the variations of kappa?*

We agree with the reviewer and we will include traces for sulfate and organics measured with the ACSM in Figure S3.

Kappa exhibits systematically lower values from 06:00 to 09:00 LT compared to the values between 12:00 and 21:00 LT, when considering the data derived from the ACSM and the CCN counter data, respectively. In particular, the increase was estimated to be as 21% when the ACSM data were considered, and 21, 24, 29, 69, and 42% for supersaturations under 0.20%, from 0.20 to 0.40%, from 0.40 to 0.50%, and from 0.60 to 0.73%, respectively when the CCN data were used.

*Comment 29: L370: Is the kappa difference of 0.2 kappa units calculated from average values? Because from Fig. S3, it seems that the difference can reach almost 0.4 (during the night and at the end of the afternoon).*

Indeed we used average values. This will be clarified in the revised version.

*Comment 30: L373-374: In connection with comment 28, the convergence which is reported in this sentence is also not obvious for me.*

We will address this by modifying the Fig. S3. A 2-4 running average (equivalent to 30/60 minutes) in the diurnal evolution of the kappa derived from the CCN clearly shows a convergence with the respective ACSM data.

*Comment 31: L374-378: This sentence is confusing me a lot, as, in my view; it conflicts with some ideas which are discussed elsewhere in the paper:*
*- "This constant character of the chemically derived kappa may be an evidence that using prescribed levels of supersaturation or critical diameters to calculate CCN concentrations can provide a biased influence of NPF events on CCN". In my view, the observation of the constant character of kappa does not indicate at this stage that the CCN predictions obtained from prescribed levels of supersaturation or prescribed diameters are biased; it only highlights the fact that both approaches are finally very similar, since the limited variations of kappa lead in the end to almost prescribed $d_c$. This assertion is even more surprising that Fig. S2 and L281-294 highlight a pretty good agreement between CCN prediction from ACSM/SMPS data and direct CCN measurements.*

The approaches may be similar, but taking prescribed diameter or supersaturation may be substantially different from those occurring in the "real" cloud-forming conditions. This is what "bias" in our discussion refers to. The results of the manuscript support this quite well.

*- "since there is a clear dependence between the chemical composition and the size of a particle": isn't is conflicting with L275-276 ("a size dependant consideration of hygroscopicity is therefore deemed unnecessary") and end of Section 2.4?*

The CCN data does indicate a size-resolved dependence in composition, but as using the bulk kappa shows, the end result in CCN error is not significant. When translated to droplet number, this error is further reduced. This is discussed in Section 3.3 of the manuscript. Given this, we feel that no further change or clarification is necessary.

*Comment 32: L386-389: The phrasing of the sentence is quite confusing; also, I wonder if it is relevant to apply this classification, which is exclusively based on measurements conducted in Pittsburgh, to measurements conducted at Finokalia, where particle concentrations and NPF event characteristics are most likely different. I would remove the sentence because I don't think it provides valuable information.*

Amended.

*Comment 33: L396: What does "intermediate nucleation mode particles" mean?*

This point was also raised by Reviewer #1, please see our relevant response there.

*Comment 34: L411: "the time series of the aerosol size distribution and chemical composition": since different datasets/instruments/measurement techniques were involved in this work, I would clearly recall that when filter measurements were used, there was only one kappa value available per day, to keep the message as clear as possible.*

We agree with the reviewer and this will be clarified in the revised version.

*Comment 35: L415: I think there is a space missing: "supersaturations" -> "supersaturations".*

Amended.

*Comment 36: L417: Information in the brackets is not useful.*

Done.

*Comment 37: L420: It might be useful to also indicate $t_{start}$ on Fig. 3a. Also, the expression "Prior to 8:30 LT and 5 hours later" is not clear to me.*

We will add the $t_{start}$ on Fig. 3a according to your suggestion. The "5 hours later" refers to the subsequent period after the start of the event, which was at 08:30 LT. This will be clarified in the revised version.

*Comment 38: L429: "dividing" instead of "diving"?*

Amended.

*Comment 39: L431: Over which period was the average value calculated? Full day?*

The average value was calculated at each time step (5-min temporal resolution). We will make this point clear in the revised version.

*Comment 40: L436-437: "from the influence of NPF on the larger supersaturations": even if I get the message, I think it would be more correct to change the wording to something like "from the influence of NPF on the production of particles which activate at larger supersaturations".*

We will follow the reviewer's suggestion and change the text accordingly.

*Comment 41: L438: On the 29th of August, the influence of NPF on CCN production is said to terminate at 21:30; was it decided that this "end time" would systematically be identified on the day of the NPF event, or did the authors extended their research period to the next day, to document growth processes possibly spanning on several days?*

We thank the reviewer for this excellent point. The truth is that we have no real means of determining the continued growth from a NPF from a previous day-beyond the point of $t_{Nd}$. Comprehensively addressing this requires a full microphysical model application and its application on a case-by-case basis, which is beyond the scope of this study. This point will be noted in the text, however.

*Comment 42: L445-446: "this variation of $R_s$ can be equivalent to the percentage contribution of CCN owing to NPF": I would suggest to change to wording to something like "This variation of $R_s$ indicates, for each supersaturation value, the increase of the CCN concentration related/due to particles originating from NPF".*

We will follow this suggestion.

*Comment 43: L453: "the time which the new particles after the $t_{start}$ are able to grow": I would suggest to rephrase this part of the sentence to make it clearer.*

This will be further clarified in the revised version.

*Comment 44: L454-455: "This time fluctuates from 2.7 to 10.5 h in the 1.0-0.38%".The value of 2.7 h was obtained assuming an initial diameter of 25 nm for the newly formed particles at $t_{start}$, is that right? If so, I am not convinced by this approach, since I would expect most of the particles in the nucleation mode to be smaller than this upper limit at $t_{start}$.*

Indeed the value of 2.7 h was obtained for particles of initial diameter of 25 nm to grow to CCN relevant sizes. This approach provides a lower limit in the time, given that additional time is required for the smallest detected particles at $t_{start}$ (9 nm).

*Also, how did the authors get the upper value of 10.5 h? For me the longest time delay should correspond to particles with a diameter of 9 nm at $t_{start}$, which then need to reach 67 nm to be able to act as CCN at s = 0.38%, i.e. + 58 nm. Considering a GR of 3.7 nm/h, I find that it takes approximatively 7 hours for the particles to reach this*

***$d_c$. Repeating the same calculation with initial diameter of 25 nm leads to a bit more than 11 hours. The hypotheses used for this calculation should be clarified.***

As suggested in the previous comment, based on the new calculations, the time intervals will be adjusted accordingly. Indeed, based on our initial calculation it will take 11.3 h for particles of 25 nm at $t_{start}$ to reach 67 nm with a growth rate of 3.7 nm h$^{-1}$. If now we consider 9 nm particles at $t_{start}$ with the same growth rate they will need 15.7 h to grow to 67 nm particles. When adding the calculations at 0.10% supersaturation which lead to a $d_c$ of 162 nm the respective time periods are 37 h for 25 nm particles at $t_{start}$ while 9 nm particles will need 41 h to reach those 162 nm. This will be clarified in the revised text.

***Comment 45: L456: "start to feel the influence of NPF". In connection to comment 40, I also get the message but I would rephrase this part of the sentence, and refer more to the time it takes for the newly formed particles to grow to $d_c$ = 67 nm.***

Done.

***Comment 46: L456-458: "$t_{dec}$ is later for supersaturations below 0.7%": I do not understand the meaning of this sentence, since based on the definition from L425-429, there is one single value of $t_{dec}$ per event, which is derived from all supersaturations. Do the authors mean that it takes longer time to observe the influence of NPF on the concentration of particles able to act as CCN at lower supersaturations, as it is expected that those need to grow to larger sizes? Also, the link with the second part of the sentence is not clear to me.***

In response to the comment raised, indeed there is one single value of $t_{dec}$ which is derived from all supersaturations using the time evolution of the $R_s$ and its relative dispersion (RD). When assuming that the GR is the solely factor determining the time delay between $t_{start}$ and $t_{dec}$, the newly formed particles require longer time to reach the critical diameters (different supersaturation level), compared to the time delay which is calculated from the $R_s$ and RD, respectively. Altogether, the $t_{dec}$ is observed earlier, compared to the time delay when using only a constant GR. This difference is due to the variability of the growth rate, which increases with an increasing particle diameter (Paasonen et al., 2018), as well as to the several microphysical processes (see below our response to your comment 53) which influence the time lag between $t_{start}$ and $t_{dec}$.

This point will be clarified in the revised version.

*Comment 47: L461-462: "indicating that the newly formed particles in this size range may exhibit similar chemical composition". Similar chemical composition at all sizes was assumed from the beginning of the calculation with the use of a single kappa for all sizes, wasn't it?! (L275).*

What we wanted to point out is that our assumption of a similar chemical composition (single kappa for all sizes) is further supported by the CCN calculations based on the similar values of the $R_s$. Nevertheless if it is considered as a circular reference it can be omitted.

*Comment 48: L464-466: How did the authors get the reported values? By averaging all $R_s$ between 13:30 and 21:30? If so, in connection to comment 42, I would again talk more about an increase of the CCN concentration due to NPF rather than a contribution of NPF, and I would suggest to further check this aspect throughout the manuscript.*

Indeed, we calculated the reported percentage perturbations using the average values of the $R_s$ between 13:30 and 21:30 LT.

This aspect will be checked in the revised text, according to your comment.

*Comment 49: L466-467: "since the $d_c$ … respectively": I do not think this is a proper explanation, I would rather say "consistent with similar $R_s$ observed in the same size range, as mentioned above".*

Done.

*Comment 50: L469-476: I am not convinced by the suggested correction process for the two main reasons which are developed below:*
*- Background particles possibly contributing to CCN population together with growing particles originating from NPF are those which were already large enough before $t_{start}/t_{dec}$, not those in the nucleation mode before $t_{start}$. And, by the way, particle concentration in the nucleation mode should be around zero before $t_{start}$, as by definition those particles originate from nucleation.*

Based on the derived critical supersaturation calculation, aerosol particles in the size range around 35 nm could contribute to CCN population at the highest level of supersaturation (1.00%). When looking at the respective diurnal number size distribution (Fig. 1a) we saw that particles in that size range also pre-existed the NPF

event ($t_{start}$) and those represent the calculated bias of up to 50% mentioned in the text. Those particles are large enough and also have enough time to grow to CCN-relevant sizes.

*- Also, if I am not mistaken, this paper discusses the CCN concentration increase from a reference concentration taken at $t_{start}$, which, I expect, already includes some contribution of background particles. I would thus say there is no need to apply any correction. The only bias, which is complex to evaluate but should still be mentioned, is caused by the possible appearance of large particles not originating from NPF between $t_{start}$ and end of NPF influence on CCN concentration (21:30 in this case), as those can impact the variations of the CCN concentration predicted at a certain (most likely low) supersaturation during this time period.*

Indeed, this study refers to the perturbations from newly formed particles on CCN number concentrations beyond the $t_{start}$, however no correction was suggested/applied. We merely mention the upper limit of bias which could originate from the pre-existence of large enough particles (not originating from the NPF) that can grow to CCN-relevant sizes. This will be further clarified in the revised text.

*Comment 51: L478: I would suggest to remove "to the $R_s$ and subsequently".*

Done.

*Comment 52: L488: "for" should be removed.*

Done.

*Comment 53: L486-491: I would have expected particle GR to be the main factor determining the time delay between $t_{start}$ and $t_{dec}$, but the seasonal variation of this time delay (similar in winter, spring and summer, and lower compared to autumn) is not consistent with that of the GR reported by Kalivitis et al. (2018) (higher GR in summer, lower in winter and spring). Could the author comment on this aspect?*

We understand the reviewer's point. According to Kalivitis et al. (2019) higher growth rates are calculated for summer and autumn (average 7.58 and 5.33 nm h$^{-1}$, respectively), while lower values during winter and spring (average 3.94 and 3.55 nm h$^{-1}$, respectively). Hence, it would be expected that the time delay between $t_{start}$ and $t_{dec}$ would be lower during summer and autumn, if only the influence of growth rate is taken into account. Nevertheless, the GR is not entirely responsible for the growth of the

freshly nucleated atmospheric particles into CCN-relevant sizes and cloud droplets (as also seen in comment 44, 46); other microphysical processes (e.g. the synoptic wind flow, the boundary layer dynamics, the presence of pre-existing particles, and the atmospheric chemical composition) favor the NPF and consequently determine the $t_{dec}$.

One should take into account that the air masses reaching at Finokalia had a substantial number of pre-existing particles which provide more surface available for condensation and coagulation. Thus, small particles which not originate from NPF are also able to grow towards larger sizes and thereby can act as CCN. Therefore, this time lag between $t_{start}$ and $t_{dec}$ could be attributed to the different amount of pre-existing particles among the seasons (e.g. during summer and autumn the number of pre-existing particles before $t_{start}$ was higher compared to winter and spring), which via the atmospheric condensation reach the CCN relevant sizes.

There is also another point which must also be addressed regarding the time delay between $t_{start}$ and $t_{dec}$, and this is associated with the strong northern wind flow which dominates upwind of Finokalia (Etesians) during summer. In particular, air masses containing a large load in Aitken mode particles and consequently consisting of already activated CCN particles are transported and end up at Finokalia owing to the advection in the MABL. This is consistent with previous studies in the vicinity of Finokalia, where during Etesian NPF events increased number concentrations were observed, with dominating Aitken mode particles (Tombrou et al., 2015; Kalkavouras et al., 2017).

***Comment 54: L494-495: "at cloud supersaturations encountered in this environment". To me, this sentence conflicts with what I think is a main message of the authors, yet already reported by Kalkavouras et al. (2017): L563, "the actual cloud supersaturation being much lower than the prescribed levels in the CCN analysis". (Also L33)***

We kindly disagree with the reviewer's comment, as the CCN analysis was carried out assuming prescribed levels regarding the supersaturation, whilst the droplet number calculations dynamically determined the supersaturations as needed in clouds.

***Comment 55: L496: Should be 3.3 instead of 3.4.***

Done.

***Comment 56: L504: I think the time interval between 8:30-11:00 is not correct to refer to the "growth hours" of the episode. This is at least not consistent with the fact***

*that the influence of growing newly formed particles on CCN population is seen between 13:30 and 21:30.*

We agree. The time interval between 08:30-11:00 LT corresponds to the formation hours. Thus, we will rephrase the sentence in the revised version, according to your suggestion.

**Comment 57: L508: Space missing between "increases" and "4.7".**

Done.

**Comment 58: L509-515: "Both trends are related to decreases in accumulation mode aerosol number": the decrease of accumulation mode particle concentration is not clearly visible on Fig. 1.**

The point is well taken. Here, we present a more detailed Figure to show the decrease of accumulation mode particle concentration from 08:30 to 17:25 LT. We will modify Figure 1c accordingly, placing in a right-hand y-axis the accumulation mode concentration.

[Figure]

*Also, I wonder why at this stage of the analysis the variations of $N_d$ are related to accumulation mode particles only, since the results of the previous section suggest a major contribution of Aitken mode particles to CCN population ($d_c < 67$ nm). To support their assumption, the authors would first need to discuss the inconsistency between the supersaturations used to predict CCN concentrations and $s_{max}$ retrieved by the model; $s_{max}$ being lower, it implies that particles need to grow to larger sizes to effectively act as CCN, and yes, in the end they most likely belong to accumulation instead of Aitken mode.*

For $\sigma_w$=0.3 m s$^{-1}$ the average $s_{max}$ between 08:30 and 17:25 LT was calculated to be 0.13%, and for $\sigma_w$=0.6 m s$^{-1}$ 0.17%, respectively. Hence, taking into account these low values of supersaturation formed in the clouds, we estimated the critical diameter ($d_c$) at 0.10% supersaturation and we found that the $d_c$ exhibits a mean value of 162 nm, indicating that most of the activated CCN belongs to the accumulation mode. Thus, the temporal fluctuations of N$_d$ converge with the respective variations of the accumulation mode aerosol number concentrations (see the above Figure and the Figure 5 in the main text). This point will be clarified in the revised text.

*≪as the latter has not had the chance to influence particles that act as CCN in clouds": again, I think this is not correct at this stage of the study, since the previous section highlights the influence of NPF on CCN concentration already from t$_{dec}$, i.e. 13:30. The supersaturation inconsistencies recalled above are also needed to further explain/clarify this aspect. Particles can act at CCN already from t$_{dec}$, but in the presence of supersaturations which are most likely significantly higher than those predicted by the model.*

*Considering the different supersaturations discussed in sections 3.2 and 3.3, I think it is finally complex to establish/comment on the link between t$_{dec}$ and t$_{Nd}$.*

We kindly disagree with the reviewer. The comment refers to the decrease in accumulation mode particles associated with processes other than NPF before the newly-formed accumulation mode particles from the NPF start to influence their concentrations after $t_{Nd}$. What we aim to point out is that when choosing prescribed levels of supersaturation in the CCN analysis the actual supersaturation levels for clouds at Finokalia calculated by the simulations are actually lower, providing biased insights regarding the influence of the NPF in clouds when solely looking at the impact on CCN concentrations. Thus the time lag between $t_{dec}$ and $t_{Nd}$ can be ascribed to those inconsistencies between the different supersaturations.

*Another point: how do the authors determine the beginning of NPF influence on N$_d$ at 17:25? From Fig.5, it seems that the most significant increase of N$_d$ is seen from ~21:00. Now looking at Fig. 1.A, this time coincides with the time at which the NPF event is somewhat interrupted, suggesting that the particles contributing to the increase of N$_d$ could finally not be related to NPF. Could the authors comment on that?*

Having calculated the supersaturation levels for clouds forming at Finokalia during the NPF episode (0.13 and 0.17% for 0.3 and 0.6 m s$^{-1}$, respectively), we found that $d_c$ displays values between 153 to 186 nm when the Kölher's equation was used for $s$ 0.10%, which suggests that the CCN particles coincide with accumulation-mode particles. From the Figure in comment 58 we may see that the number concentration of particles in accumulation-mode shows a minimum value at 17:25 LT, and this time stamp coincides with the $t_{Nd}$.

Furthermore, the influence of the NPF on droplet number is said to start at the time when the $N_d$ exhibit minimum values after $t_{dec}$, and when the maximum supersaturation starts to decrease at the respective time, since the CCN particles which have already been formed start to grow further (competing for water vapor thus decreasing $s_{max}$) to form droplets.

*Comment 59: L519-529: It is complex for me to understand this part of the analysis. Specific comment on L527, "Since $N_d$ does not increase significantly until midnight": looking at Fig. 5 I would say it does, at least until 23:00. Second part of the sentence is also not clear to me.*

We apologize for this. What is meant is that the influence on $N_d$ is not seen but until very late in the evening. This will be revised in the text.

*Comment 60: L533: I do not understand how the authors calculate the value of 30 cm$^{-3}$.*

30 cm$^{-3}$ is the average value of the variance of the droplet number as described in Section 2.5, from 17:25 until midnight when $\sigma_w$=0.3 m s$^{-1}$, and the 35 cm$^{-3}$ is the corresponding value when doubling the $\sigma_w$. We will clarify this.

*Comment 61: L534: In connection to comment 41, why is the analysis stopped at midnight? Again, if the analysis is limited to the day of the event, this has to be mentioned, also the reason why.*

Done. We will clarify this point in the revised text according to your suggestion.

*Comment 62: L535-538: I think the length of the dataset is a strong point of this work, so I would really suggest to discuss more the "statistics" in the main text. In general, the supplementary includes valuable information, from which the reader would benefit more if it was partly moved to the main text. This comment also applies*

*to CCN related calculations reported in the previous section (see for instance comments 9 and 23).*

These are excellent points! We will try to include a section in order to discuss more the "statistics" regarding the seasonal variability in the revised main text.

*Comment 63: L548: "accurately" is in my view too strong.*

Amended.

*Comment 64: L550, 553: time delays between $t_{start}$ and $t_{dec}$ are different from those reported in Sect. 3.2. Also, it would be better to have a uniform notation to report durations (L550, 553 and 554).*

Amended.

*Comment 65: L558-559: "the impact of NPF on $N_d$ differs considerably from the CCN based analysis". I do not completely agree with this sentence, as it compares two different variables (CCN concentration and $N_d$) calculated using different hypotheses.*

We kindly disagree here. Droplet number is equal to the CCN at the cloud-relevant supersaturation. For this reason, the $N_d$-based analysis differs from the CCN-based because the supersaturation is much lower and variable. This is one of the most important aspects of the paper, so no changes are made.

*Comment 66: L559: "Regardless of season". This assessment is a bit too strong in my view, as a possible seasonal variation of some parameters such as vertical velocity (and thus further effect on $N_d$ calculation) has not been discussed. Also, I would not talk about "typical boundary layer clouds"; L310 indicates "in cloudy boundary layer in the region".*

Reviewer #1 raised a similar point; please see our discussion there regarding the issue of updraft velocity and its seasonality.

This study refers to the Eastern Mediterranean, so the typical boundary layer cloud in our discussion refers to that found in the region.

*Comment 67: L569-570: Please refer to comment 50.*

This has already been addressed.

*Comment 68: L571-575: I would not say this is a striking consequence; I would rather say it is expected, particles need to grow to large-enough sizes to be activated into droplets, which takes time, and this is even more the case when supersaturations get lower. I am also not sure about the last part of the first sentence "$N_d$ is insensitive to increase in CCN during the course of an event… vapor": again, during the course of the event, CCN population possibly activating at higher supersaturation is increased first, and as particles are getting bigger towards the end of the event, they can activate at lower supersaturations.*

It is seldomly specified in the literature how late in the day the NPF impacts on droplet number can be. This is striking, given how persistent and consistent the timings seem to be.

The insensitivity reflects that a very small fraction of the total CCN perturbations seems to manifest in droplet number. No changes are deemed necessary.

*Comment 69: L579-580: Wasn't it already a result from Kalkavouras et al. (2017)? See also comments 9 and 23.*

The study of Kalkavouras et al. (2017) refers to a specific event, across two sites during summertime. This study does not focus on the special extent of NPF and spans a period of seven years. Unlike Kalkavouras et al., conclusions presented here are much more general and relevant for the regional climate.

*Comment 70: L581: The influence of what?*

We apologize for this, and we refer to the influence of the chemical composition and aerosol number during NPF events. We are going to modify the sentence according to your suggestion.

*Comment 71: L584: "highly effective paradigm" is too strong for me; if new metrics are introduced for quantifying the temporal aspect of NPF influence on CCN production, CCN calculation itself relies on the use of prescribed supersaturation, which is simultaneously reported to be hazardous (eg L525, 581).*

We will modify it in the revised version according to your suggestion.

*Comment 72: L741-749: Check alphabetical order.*

Done.

*Comment 73: Fig. 1: Caption, (b), I would remove "differences", and simply refer to the time series of the particle concentration at different time resolutions.*

Done.

*Comment 74: Fig. 3: it would help to have similar scale on the x axis of Fig. 3a and b.*

As Fig. 3a refers to $R_s$, while Fig. 3b to the relative dispersion of $R_s$ we feel that using similar scale is inappropriate.

*Comment 75: Fig. S1: x axis label: "ACSM" instead of "ASCM". Also, adding the 1:1 line would help to interpret the results.*

Done.

*Comment 76: Fig. S3: Caption: space missing between "the" and "kappa".*

Done.

*Comment 77: Table S2: I would suggest to clearly explain what "bef" and "aft" refer to, as these are used in several tables of the supplementary. Also, in the caption, it is not clear to which "relative contribution" the authors are referring to. I would clearly mention it is the increase of $R_s$ (%) observed after $t_{dec}$, and clearly indicate it corresponds to the "Change" column.*

Done.

*Comment 78: Table S5: If I am not mistaken, the authors never refer to Table S5, neither in the main text nor in the supplementary. This table report values which would have been very interesting to discuss more in the main text, in specific those calculated for s=0.1%, as this supersaturation is thought to be more "representative" of real clouds. Also, more information would be needed regarding the calculation of the $N_{CCN}$ values, as I think it has not been explained elsewhere: over which time period (average between $t_{dec}$ and end of NPF influence?)? Any correction applied?*

Indeed so, we never refer to the Table S5, thus we will add more information in the main text (especially the calculations for supersaturation 0.10%) according to your suggestion.

*Comment 79: Supplementary: several spaces missing in different places.*

Done.

**References**

[revised manuscript text omitted]